 

# A balance of positive and negative regulators determines the pace of the segmentation clock

**Guy Wiedermann[1†], Robert Alexander Bone[1†], Joana Clara Silva[1], Mia Bjorklund[1], Philip J Murray[2‡], J Kim Dale[1*‡]**

[1]Division of Cell and Developmental Biology, College of Life Sciences, University of Dundee, Dundee, United Kingdom; [2]Division of Mathematics, University of Dundee, Dundee, United Kingdom

**Abstract** Somitogenesis is regulated by a molecular oscillator that drives dynamic gene expression within the pre-somitic mesoderm. Previous mathematical models of the somitogenesis clock that invoke the mechanism of delayed negative feedback predict that its oscillation period depends on the sum of delays inherent to negative-feedback loops and inhibitor half-lives. We develop a mathematical model that explores the possibility that positive feedback also plays a role in determining the period of clock oscillations. The model predicts that increasing the half-life of the positive regulator, Notch intracellular domain (NICD), can lead to elevated NICD levels and an increase in the oscillation period. To test this hypothesis, we investigate a phenotype induced by various small molecule inhibitors in which the clock is slowed. We observe elevated levels and a prolonged half-life of NICD. Reducing NICD production rescues these effects. These data provide the first indication that tight control of the turnover of positive as well as negative regulators of the clock determines its periodicity.

*For correspondence: j.k.dale@dundee.ac.uk

[†]These authors contributed equally to this work

[‡]These authors also contributed equally to this work

**Competing interests:** The authors declare that no competing interests exist.

## Introduction

The formation of the vertebrate skeleton and associated musculature relies on the ordered and timely segmentation of mesodermal tissue during early embryonic development. Segmentation is progressive and occurs in an anterior to posterior direction as the body axis elongates (reviewed in *Oates et al., 2012*). Mesodermal segments, or somites, 'pinch off' at the rostral limit of the two rods of pre-somitic mesoderm (PSM) that lie on either side of the caudal neural tube, with a distinct, species-specific periodicity; every 90 min in the chicken and every 120 min in the mouse. This periodicity is regulated by a molecular oscillator, known as the segmentation clock, that is defined by cyclic gene expression within PSM tissue with a period that regulates somite formation. These 'clock' genes belong to the Notch, WNT and FGF signalling pathways and are expressed in a cyclical pattern in the PSM of a wide variety of vertebrate species (*Krol et al., 2011*; reviewed in *Dequéant et al., 2008*; *Gibb et al., 2010*; *Maroto et al., 2012*). On a single cell level in the vertebrate PSM, oscillatory gene expression is believed to be established through negative feedback loops of unstable clock gene products.

Previous mathematical models of the somitogenesis clock that invoke the mechanism of delayed negative feedback predict that the clock period can be approximated as a sum of the delays involved in processes such as transcription, splicing, translation and transport and the half-lives of inhibitors of clock gene expression (*Lewis, 2003*; *Monk, 2003*; *Ay et al., 2013*; *Hanisch et al., 2013*). Subsequent studies have confirmed that the period can be altered by genetically modifying genes that encode negative regulators (e.g., Hes7, Her6, Her7, Nrarp) (*Herrgen et al., 2010*; *Schröter and Oates, 2010*; *Kim et al., 2011*; *Takashima et al., 2011*; *Oates et al., 2012*; *Harima, et al., 2013*; *Hoyle and Ish-Horowicz, 2013*).

**eLife digest** During embryo development, animals with backbones (also called vertebrates) repeatedly lay down pairs of segments along the axis that runs from the head to the tail of the embryo. These segments, known as somites, eventually form part of the skeleton, as well as the associated muscle, cartilage, tendons and some skin. Importantly, the segments in some species take longer to form than those in other species, and they also form in proportion to the overall size of the animal.

A 'segmentation clock' regulates the timing of somite formation via cycles in which genes are repeatedly switched on and then off again. Some aspects of this process are well understood. Firstly, many 'clock genes' are known to produce proteins that can inhibit their own production. However, this 'negative feedback' is typically delayed because it takes time to produce and transport protein within a cell. The inhibitory proteins are also unstable and their breakdown leads to an end of their inhibitiory effect. It is also known that: some proteins send signals to neighbouring cells while others, including one called Notch, receive them; and the received signals activate the expression of clock genes. However, until now, no one had studied how the turnover (that is, the production and breakdown) of the proteins that activate clock gene expression could regulate the pace of the clock.

Wiedermann, Bone et al. used a two-pronged approach to investigate this question. First, they developed a computational model that accounted for both inhibition and activation of clock gene expression. The model predicts that the clock slows down when the levels of a positive regulator called Notch intracellular domain (or NICD for short) are high. This is because the negative regulators would have to overcome the increased positive regulators to switch off the clock genes. A slower segmentation clock would be expected to give rise to fewer, larger somites in a given length of time when compared to a similar clock with a faster pace.

To test these predictions, Wiedermann, Bone et al. next conducted experiments on chicken embryos, which are commonly used in studies of animal development. The experiments agreed with the model predictions. That is, when treated with a variety of drugs that affected NICD turnover and thereby increased the levels of NICD, the clock slowed and these chicken embryos developed fewer, but larger somites. As predicted by the mathematical model, these effects were rescued when Wiedermann, Bone et al. reduced the production of NICD. These findings show that a balance of positive and negative regulators determines the pace of the segmentation clock.

Moreover, it has been shown that regulation of clock gene mRNA turnover and degradation is gene specific and regulated at the level of the 3′UTR (*Hilgers et al., 2005*; *Nitanda et al., 2014*). In contrast, there has been relatively little experimental work demonstrating the predicted role of protein stability in regulating the periodicity of clock gene oscillations (*Hirata et al., 2004*).

It is notable that models that invoke delayed negative feedback as the mechanism underlying the somitogenesis clock were developed primarily from zebrafish data, where the clock period is relatively short and there are significantly fewer oscillating components than other vertebrate species (*Krol et al., 2011*). Moreover, in zebrafish there is experimental evidence that Notch plays a key role in the synchronisation of neighbouring oscillators (*Delaune et al., 2012*) but that it is not necessary for oscillations themselves (*Ozbudak and Lewis, 2008*). In contrast, Notch signalling in mouse and chick is thought to be essential for oscillations (*Ferjentsik et al., 2009*). One crucial factor in this regard is the Notch1 intracellular domain (NICD) which is cleaved following ligand activation, translocates to the nucleus and activates Notch target gene expression. NICD is unstable and, at least in mouse, its production appears as pulsatile, spatio-temporal waves that traverse the rostro-caudal axis of the PSM in the manner of a clock gene (*Huppert et al., 2005*). Importantly, pulsatile Notch1 protein expression was recently shown to be dependent on Notch signalling in the PSM of both chick and mouse (*Bone et al., 2014*); this raises the possibility that, in mouse and chick at least, positive feedback, mediated by Notch and NICD, plays a role in the clock mechanism.

In this study we develop a mathematical model of the clock that describes both repression and NICD-mediated activation of clock gene transcription. The model predicts that increasing the half-life of NICD results in elevated levels of NICD and a longer clock period. Notably, these effects are not observed in the model upon increasing the half-life of the repressing clock protein. To test the

predictions, we use a pharmacological approach that robustly delays the phase patterns of a clock gene in the chick and mouse PSM. Under these conditions, using a custom made antibody that recognises endogenous chick NICD, we show that, concomitant to delaying the pace of oscillations, the inhibitors increase both levels of NICD and its half-life. In agreement with model predictions, we rescue the phase patterns of clock gene expression by reducing NICD production. These data therefore suggest that increasing the half-life of NICD and thus increasing its turnover time increases the period of the somitogenesis clock.

## Results

### Mathematical modelling predicts that increased NICD half-life results in higher levels of NICD and a longer clock period

In order to investigate the role of both positive and negative feedback in the circuitry of the somitogenesis clock, we developed a mathematical model in which the key regulatory interactions are that an inhibitory clock protein and NICD negatively and positively regulate gene transcription, respectively (see 'Materials and methods' and *Figure 1A,B*). Hence in the absence of clock protein but presence of NICD, transcription occurs. Conversely, transcription is inhibited when there are high levels of inhibitor and low levels of activator.

In *Figure 1C* we plot a representative numerical solution of the model. In a given cycle of the clock, NICD activates transcription, resulting in the production of cytoplasmic protein after some delay, $T_1$. Clock protein then represses transcription and after a further delay, $T_2$, NICD is produced and activates the next cycle of the clock. In the model, perturbing protein half-lives alters the balance between activation and repression in a manner that modifies the clock period. In *Figure 1D* we show that increasing the NICD half-life increases the period of the oscillation and the levels of both NICD and clock protein. In *Figure 1E* we show that increasing the clock protein half-life does not greatly affect the oscillator period but reduces levels of NICD.

In order to further explore model behaviour we calculated solutions of the model in different regions of parameter space. In *Figure 1F,G* we show that the oscillator period and levels of NICD increase with NICD half-life but that these effects are not seen upon increasing the clock protein half-life. In *Figure 1H* we show that the increased period, arising as a consequence of increased NICD half-life, can be rescued by reducing the NICD production rate. These numerical solutions demonstrate that, in the model, the balance between NICD production and degradation rates plays a key role in determining the clock period. Additionally, we note that there is a critical value of the production rate below which the trivial steady-state becomes stable (solutions tend to zero) and that perturbing clock protein and NICD production and decay rates can drive the system into stable states (e.g., by increasing repressor half-life in *Figure 1F* or reducing the NICD production rate in *Figure 1H*).

### Wnt inhibition delays the pace of *cLfng* oscillations in the chick and mouse PSM

It has previously been reported that modulating either the Shh or Wnt pathways can affect the pace of clock gene oscillations in the PSM (*Gibb et al., 2009*; *Resende et al., 2010*; *Gonzalez et al., 2013*). To experimentally test the predictions of our model, we used a half-embryo assay to investigate whether levels and half-life of NICD were elevated under conditions where clock gene oscillations were robustly delayed using a pharmacological approach (see 'Materials and methods'; *Gibb et al., 2009*). In the first instance, PSM explants from chick embryos between Hamburger Hamilton (HH) stages 10–12 were exposed to XAV939, a specific Wnt inhibitor that acts by inhibiting Tankyrase1 (TNK1) and TNK2 enzymes, which normally degrade Axin2. Hence upon treatment with XAV939, AXIN2 protein is stabilised and maintains phosphorylated β-catenin protein in the destruction complex (*Huang et al., 2009*).

Following a titration assay, we determined that 100 μM XAV939 treatment was the lowest concentration that caused robust and reproducible down regulation of the Wnt target genes *cAxin2* and *cLef1* compared to the corresponding DMSO-treated contralateral explants (*Figure 2—figure supplement 1C,D*; n = 61/68; *Bone et al., 2014*) and led to increased levels of phosphorylated β-catenin at Ser33, Ser37, Thr41, as expected (*Figure 2—figure supplement 1A*; n = 8/10). Using the same assay, we investigated the effect of the inhibitor on the dynamic

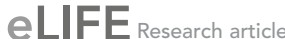

**Figure 1**. A mathematical model of positive and negative regulation of the somitogenesis clock. (**A**) A schematic illustration of inter-cellular coupling via Delta-Notch signalling. (**B**) A schematic illustration of the reduced model in which the pulsatile production of NICD is down stream of clock gene expression. (**C–E**) Representative numerical solution of the model *Equations (1)-(3)*. Clock protein (p(t), solid line) and NICD (n(t), dashed line) levels are plotted against time. (**C**) 'Wild-type' oscillations ($k_4 = 0.063$), (**D**) Decreasing NICD decay rate yields longer period oscillations ($k_4 = 0.023$), (**E**) Decreasing Hes7 decay rate lowers levels of Hes7 and NICD ($k_2 = 0.035$). (**F**) The oscillation period (colour) is plotted against Hes7 and NICD half-lives. (**G**) NICD level is plotted against Hes7 and NICD half-lives. (**H**) Oscillator period is plotted against NICD production rate and half-life. (**E–G**) Solid lines

*Figure 1. continued on next page*

Figure 1. Continued

depict emergence of the non-trivial steady-state. Dashed lines depict points in parameter space where the nontrivial steady state undergoes a Hopf bifurcation. Damped Oscillations (DO), Stable Oscillations (SO), No Oscillations (NO). Unless otherwise stated, parameter values are: $k_1$=123 min$^{-1}$, $k_2$=0.058 min$^{-1}$, $k_3$=134 min$^{-1}$, $k_4$=0.063 min$^{-1}$, $T_1$=44.0 min, $T_2$=67.0 min, $K_P$=125, $K_N$=125.

expression of *cLfng* in the PSM. Following exposure to 100 μM XAV939, the domain of *cLfng* expression was at least 1 phase behind that of the control explant (*Figure 2A*, M; n = 57/64) and occasionally this resulted in the treated explant forming one less somite boundary than the control explant. These data imply that 100 μM XAV939 treatment noticeably lengthens the period of the oscillations. In explant pairs where both sides were DMSO-treated, an asymmetric pattern of *cLfng* expression was rarely seen (n = 2/18, data not shown) and, as such, the effects of XAV939 on *cLfng* expression could not be attributed to DMSO, natural variability in expression, or to the assay itself. In order to ensure that the oscillations were delayed and not halted, a fix and culture assay (see 'Materials and methods') revealed that *cLfng* expression was still dynamic in the presence of 100 μM XAV939 (*Figure 2B*; n = 8/9). Furthermore, Phospho-histone H3 (pH3) and NucView analyses demonstrated that neither proliferation nor apoptosis, respectively, were significantly affected following drug treatment (*Figure 2—figure supplement 2A,D*; n = 4, p = 0.236; n = 4, p = 0.292). These data clearly demonstrate that Wnt inhibition delays the period of the segmentation clock in the chick PSM. Using the same half-embryo inhibitor assay in the mouse embryo, we found that exposure to 100 μM XAV939 delayed the pace of *mLfng* oscillations as compared to control DMSO-treated E10.5 half PSM explants (*Figure 2I*; n = 19/26). Moreover, a fix and culture assay revealed that *mLfng* expression was still dynamic in the presence of 100 μM XAV939 (*Figure 2J*; n = 12/20).

## Shh inhibition does not delay the pace of *cLfng* oscillations in the chick PSM

Similarly, we directly perturbed the Shh pathway in the half-embryo assay using Cyclopamine, a well-established inhibitor of Shh signalling. At 25 μM, Cyclopamine abolished expression of the Shh target gene *cGli1* in the chicken half-PSM explant compared to the DMSO control (*Figure 3A*; n = 5/5). Surprisingly, and in contrast to published data (*Resende et al., 2010*), *cLfng* expression in the PSM was not delayed at either 25 μM (*Figure 3B,D*; n = 6/9) or 50 μM (*Figure 3C,D*; n = 3/4). This assay suggests that Shh signalling does not play a direct role in the regulation of the segmentation clock.

## Exposure to XAV939 increases the level of cNICD in the chick and mouse PSM

Given our results with the Wnt inhibitor and that NICD stability is known, in other contexts, to be regulated by GSK3B (*Espinosa et al., 2003*; *Jin et al., 2009*; see also *Foltz et al., 2002*), we investigated whether the effect upon the period was associated with a change in levels of NICD, as the model predicted. To that end we generated a polyclonal anti-cNICD antibody raised against the chicken N-terminal sequence of the cleaved chicken Notch1 intracellular domain. The epitope is only exposed after gamma secretase cleavage and is not accessible in the un-cleaved form. After confirming specificity by Western Blotting against recombinant cNICD protein (*Figure 4A*, lanes 1, 2), we cultured half-PSM explant pairs in the presence of the γ-secretase inhibitor LY411575 (*Lanz et al., 2004*) or DMSO for 3 hr. Western Blot revealed a specific band against endogenous cNICD that ran at the same size as recombinant cNICD protein and at the predicted molecular weight of ~87 kDa in the DMSO control pool which was abolished in the +LY411575 treated pool, thereby validating the specificity of the antibody (*Figure 4A*, lanes 3, 4; *Figure 4D*, lanes 1, 2). We performed immunohistochemistry on sagittal sections of chick PSM tissue to analyse the spatial localisation profile of NICD in this tissue and found it to exhibit a very similar profile to that described for mNICD in the mouse PSM (*Huppert et al., 2005*). Thus cNICD is produced in (and cleared from) discrete domains of the PSM in a manner reminiscent of the expression profile of clock genes in different phases of the cycle (*Figure 4B*).

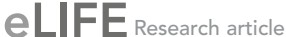

**Figure 2**. XAV939, Roscovitine, DRB and PHA767491 treatment delays the pace of the segmentation clock in the chick and mouse PSM. Bissected chick or mouse caudal explant pairs treated '−' or '+' inhibitor (**A**, **C**, **E**, **G**) or treated with inhibitor and subjected to the fix and culture assay (**B**, **D**, **F**, **H**) and then analysed by in situ hybridisation for *Lfng* mRNA expression : (**A**, **C**, **E**, **G**) Treatment of chick PSM explants in the presence (+) or absence (−) of XAV939 (**A**), Roscovitine (**C**), DRB (**E**) and PHA7667491 (**G**) for 3 hr reveals that '+' explants have lagging expression of *cLfng*, often with one less somite
*Figure 2. continued on next page*

*Figure 2. Continued*

formed than the '−' explants. (**B**, **D**, **F**, **H**): After 3 hr treatment in the presence of XAV393 (**B**), Roscovitine (**D**), DRB (**F**) and PHA767491 (**H**), one chick PSM explant was fixed while the other was treated for another 45 min, showing that *cLfng* expression is still dynamic in the presence of these inhibitors. (**I**, **K**): Mouse PSM explants treated in the presence or absence of XAV939 (**I**) or Roscovitine (**K**) for 4 hr revealed a delay in the oscillations of *mLfng* expression. (**J**, **L**): Treatment of one mouse PSM explant for 4 hr, and the other for 5 hr in the presence of XAV939 (**J**) or Roscovitine (**L**) reveals that *mLfng* mRNA expression is still oscillating in the PSM. The red arrowheads identify the somites that have formed during the in vitro culture period of the assay. (**M**) Schematic representation of the expression domains of *Lfng* in the PSM in the three different phases of one oscillation cycle. S1, SII = the most recently formed somite. (P) = previous cycle.

The following figure supplements are available for figure 2:

**Figure supplement 1**. XAV939, Roscovitine, DRB treatment down-regulates expression of the Wnt target Axin2 and the Shh target Gli1 in the chick PSM.

**Figure supplement 2**. XAV939, Roscovitine, DRB treatment does not induce apoptosis or affect cell proliferation in the chick PSM.

**Figure supplement 3**. XAV939, Roscovitine, DRB and PHA-767491 treatment delays dynamic mRNA expression of Notch target clock genes in the chick PSM thereby extending the clock period.

To investigate if exposure to XAV939 affects the levels of endogenous cNICD, chick embryos were bissected as before and contralateral sides pooled. For each reagent, one pool was cultured for 3 hr in DMSO and the other was cultured in the inhibitor. In order to cease production of new NICD, both control and inhibitor treated pools were cultured for an additional hour in the presence of cycloheximide (*Figure 4C*). Lysates were then analysed by Western Blot for NICD. Strikingly, we observed that the level of cNICD protein was increased significantly in the presence of XAV939 (*Figure 4D*, lanes 5, 6; n = 7/9; *Table 1*) compared to the corresponding DMSO control pool where the resident cNICD had degraded substantially in the presence of cycloheximide. For each assay, additional control explants confirmed that: i) *Lfng* was delayed in inhibitor treated explants; and ii) samples cultured for 3 hr in the absence of inhibitor and then for the last hour in cycloheximide alone showed a reduction in levels of NICD when compared to contralateral explants cultured in Ethanol for the last hour (*Figure 4D*, lanes 3, 4). Notably, even in the absence of cycloheximide (*Figure 4—figure supplement 1A*), levels of cNICD in the chicken PSM were consistently higher when treated with XAV939 (*Figure 4—figure supplement 1B*; n = 5/5; *Table 1*) compared to respective DMSO control lysates, although the mean fold-change increase was slightly less. These assays reveal that exposure to XAV939 elevates the level of cNICD protein in the chicken PSM, just as it delays oscillations of the Notch target gene *cLfng*. Significantly, exposure to Cyclopamine did not affect the levels of cNICD in treated PSM explants just as it did not affect the pace of *cLfng* oscillations in this tissue (*Figure 4D*, lanes 13, 14; n = 2/2; *Table 1*).

Given the consistent effect of 100 μM XAV939 upon both *cLfng* and *mLfng* oscillations, we used the same assay to investigate whether this reagent also affected levels of mNICD in the mouse PSM, with the initial culture period extended to 4 hr to allow for the longer clock period in the mouse embryo. Strikingly, we observed that the level of mNICD protein in the PSM was increased significantly in the presence of XAV939 (*Figure 4E*, lanes 3, 4; n = 8) compared to the corresponding DMSO control pool of half PSM explants, where mNICD levels had dropped substantially in the presence of cycloheximide (*Figure 4E*, lanes 1, 2; *Table 1*). Thus, the effect of XAV939 to delay the pace of *Lfng* oscillations across the PSM and to increase the level of NICD in the PSM is conserved in both chick and mouse.

## Cyclin dependent kinase inhibitors delay the pace of *Lfng* oscillations in the PSM

Previous studies have shown that the stability of NICD in other contexts can be regulated by cyclin dependent kinase (CDK) mediated phosphorylation (*Fryer et al., 2004*). Given the prediction of the mathematical model that modification of NICD stability can influence the period of the clock, we investigated whether perturbing CDK signalling would affect either clock gene oscillations and/or levels of NICD in the chick PSM. To that end, we directly perturbed the CDK pathway in the half-embryo assay using the CDK inhibitor Roscovitine. Roscovitine is a 2,6,9-substituted purine analogue that competes with ATP for the active binding site on CDKs (*MacCallum et al., 2005*).

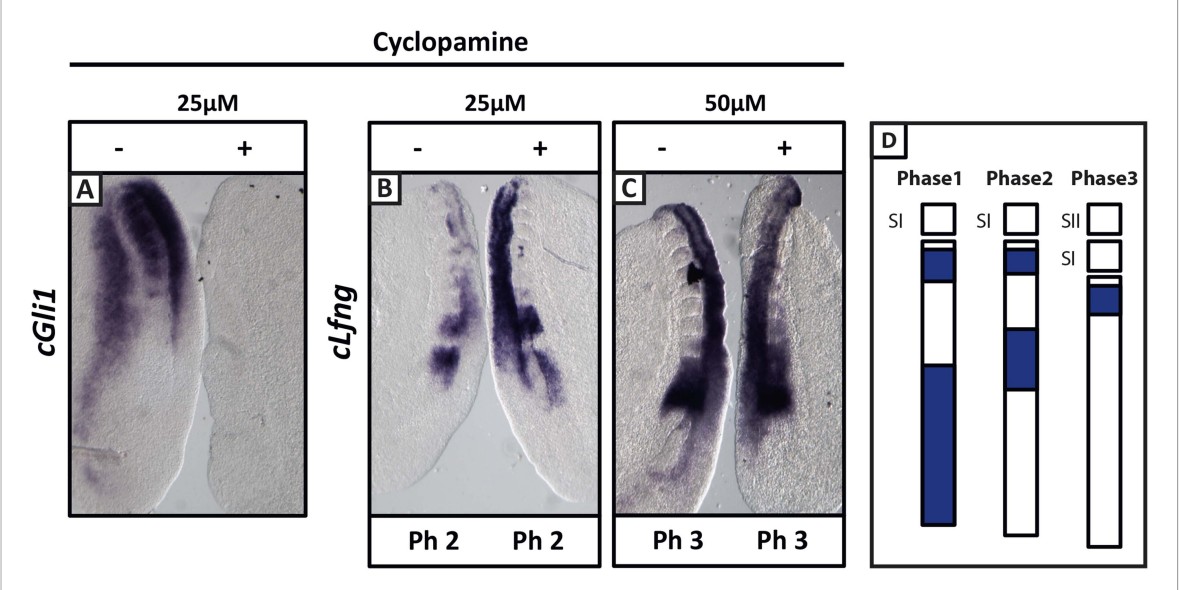

**Figure 3**. Cyclopamine treatment does not affect the pace of *cLfng* oscillations in the chick PSM. Bissected chick caudal explant pairs treated '−'or '+' cyclopamine and then analysed by in situ hybridisation for *Lfng* or *Gli1* mRNA expression. (**A**): At 25 µM, cyclopamine inhibits expression of the Shh target gene *cGli1* in the chick PSM after 3 hr treatment. (**B**, **C**): The expression domains and intensity of *cLfng* mRNA are the same in the PSM of the '−' or '+' cyclopamine halves of each embryo and there is no difference in the number of somites in each half of a pair thus we conclude the pace of *Lfng* oscillation is not affected by cyclopamine treatment for 3 hr at either 25 µM (**B**) or 50 µM (**C**). (**D**) Schematic representation of the expression domains of *Lfng* in the PSM in the three different phases of one oscillation cycle. S1, SII = the most recently formed somite.

We found that Roscovitine treatment at 10 µM clearly delayed oscillation of *cLfng* expression relative to the DMSO control (**Figure 2C**; n = 59/62) whilst the fix and culture assay confirmed that *cLfng* expression was still oscillatory (**Figure 2D**; n = 8/10). Neither proliferation (**Figure 2—figure supplement 2B**; n = 5, p = 0.204) nor apoptosis (**Figure 2—figure supplement 2E**; n = 4, p = 0.886) were significantly affected in the PSM. However, exposure to 100 or 300 µM Roscovitine severely impacted both proliferation and apoptosis and strongly down-regulated or abolished *cLfng* expression (data not shown). These findings reveal that the oscillations of *cLfng* expression in the chicken PSM are slowed by Roscovitine treatment at a concentration that does not appear to affect cell proliferation. Moreover, we found that exposure to 10 µM Roscovitine delayed the pace of *mLfng* oscillations as compared to control DMSO-treated E10.5 mouse half PSM explants (**Figure 2K**; n = 20/22). The fix and culture assay revealed that *mLfng* expression was still dynamic in the presence of 10 µM Roscovitine (**Figure 2L**; n = 8/10).

Roscovitine is a broad spectrum CDK inhibitor. We used an additional two CDK inhibitors reported to specifically reduce the kinase activities of CDK7/CyclinH and CDK9/CyclinT towards the C-terminal domain (CTD) of RNA pol II: 5,6-Dichloro-1-beta-D-ribofuranosylbenzimidazole (DRB; a nucleoside analogue) and PHA-767491. We found that exposure of chick half-PSM explants to 10 µM DRB delayed *cLfng* oscillation phase compared to the control explant (**Figure 2E**; n = 67/73) and the fix and culture assay confirmed that *cLfng* expression was still oscillatory in the PSM at this concentration (**Figure 2F**; n = 15/18). Western Blotting analysis confirmed that phosphorylation of Serine 5 on the CTD of RNA pol II was reduced by 10 µM DRB treatment (**Figure 2—figure supplement 1B**; n = 3/3; mean fold change = 0.709). Neither proliferation (**Figure 2—figure supplement 2C**; n = 5, p = 0.884) nor apoptosis (**Figure 2—figure supplement 2F**; n = 3, p = 0.844) were significantly affected. Similarly, exposure to 1 µM PHA-767491 reduced phosphorylation of Serine 5 on the CTD of RNA pol II (**Figure 2—figure supplement 1B**; n = 2; mean fold change = 0.709), delayed *cLfng* oscillation phase compared to the control half-PSM (**Figure 2G**; n = 13) and the fix and culture assay confirmed that *cLfng* expression was still oscillatory in the PSM at this concentration (**Figure 2H**; n = 6). Another cycling Notch target, *cHairy2*, was also consistently delayed compared to the respective DMSO control when treated with either XAV939 (**Figure 2—figure supplement 3A**; n = 4/6), Roscovitine

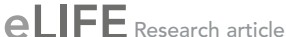

**Figure 4**. cNICD levels in the chick PSM are elevated after exposure to XAV939, Roscovitine, DRB or PHA767491.
(**A**): A polyclonal cNICD antibody was raised against the N-term sequence of the cleaved chicken Notch1
intracellular domain. The epitope is only exposed after gamma secretase cleavage and is not accessible in the
uncleaved form. By Western Blot analysis the antibody detects a band of protein at around 90KDa (see arrow) in an
*in vitro*-generated recombinant cNICD sample, which is not detected in the control IVT sample; this band
*Figure 4. continued on next page*

*Figure 4. Continued*

disappears in half chick PSM pools (see 'Materials and methods') after 3 hr treatment with 150 nM LY411575, a concentration shown previously to have no toxic effects to PSM tissue (*Bone et al., 2014*). (**B**): Immunohisto-chemistry for cNICD protein on 16 μm sagittal sections of sucrose agar embedded HH10 chicken embryo tails showing dynamic phases of localisation in the PSM and schematic representations to the right of each panel. S1 - S3 = somite; S0 - S-1 = prospective somite region of anterior PSM. (**C**): A schematic illustrating the cNICD degradation assay: corresponding pools of 9 PSM explants are incubated in the presence or absence of an inhibitor for 3 (chick) or 4 hr (mouse), before treating all pools with the protein synthesis inhibitor cycloheximide (CHX) for a further hour. Lysates are then collected for Western Blot analysis. (**D**): Representative Western Blot showing levels of cNICD protein in lysates of chick PSM pools are increased after treatment with XAV939, Roscovitine, DRB and PHA767491, but not with cyclopamine. Lanes 1 and 2 show exposure to 150 nM LY411575 alone for 3 hr removes all NICD and serves as a control for the western. Lanes 3 and 4 show exposure to CHX alone in the last hour of culture severely depletes NICD levels as compared to NICD levels in the pool of contralateral PSM explants cultured in DMSO. (**E**): Levels of mNICD in the mouse PSM pools are increased following treatment with XAV939. Lanes 1 and 2 show that NICD protein levels drop drastically after exposure for 1h to CHX as compared to strong NICD signal in lysate from the contralateral half PSMs treated with EtOH. (**F**): Representative Western Blot showing levels of cNICD protein in lysates of chick PSM pools are increased after treatment with XAV939, Roscovitine and DRB when LY411575 is added to the culture for the last hour (in place of CHX). Lanes 1 and 2 show exposure to LY411575 alone only in the last hour severely depletes NICD levels as compared to the NICD levels in the pool of contralateral PSM explants cultured in DMSO.

The following figure supplement is available for figure 4:

**Figure supplement 1**. XAV939, Roscovitine, and DRB treatment increases the level of cNICD in the chick PSM.

(*Figure 2—figure supplement 3B*; n = 6/7), DRB (*Figure 2—figure supplement 3C*; n = 5/6) or PHA-767491 (n = 8/8; *Figure 2—figure supplement 3D*). These data demonstrate that a panel of CDK inhibitors robustly delays Notch target clock gene oscillations in the PSM in a similar way to that seen following Wnt inhibition and that these effects are separable from the cell cycle.

## The CDK inhibitors do not appear to affect clock period via inhibition of canonical Wnt signalling

In order to investigate whether the CDK inhibitors all act through the Wnt pathway to elicit their effect on the clock period, we assayed transcription of the Wnt target *cAxin2* in the caudal chicken PSM. We found that 100 μM XAV939 (*Figure 2—figure supplement 1D*), 10 μM Roscovitine (*Figure 2—figure supplement 1E*; n = 31/41) and 10 μM DRB (*Figure 2—figure supplement 1F*; n = 29/42) robustly down-regulated transcription of *cAxin2* in the caudal chicken PSM, compared to the DMSO control. These data raise the possibility that the effects of these reagents on the segmentation clock are via inhibition of Wnt activity. However, the increased phosphorylation of β-catenin at Ser33, Ser37, Thr41 observed with XAV939, which serves as a readout of Wnt inhibition, was not observed following exposure to 10 μM Roscovitine (*Figure 2—figure supplement 1A*; n = 5/6) or 10 μM DRB (*Figure 2—figure supplement 1A*; n = 4/4), suggesting that the mechanism by which these CDK inhibitors inhibit transcription of *cAxin2* is not through the inhibition of canonical Wnt signalling. However, alternative mechanisms of Wnt inhibition exist that do not lead to an increase in β-catenin phosphorylation and thus we cannot rule out an effect of CDK inhibition via changes in Wnt signalling.

## Exposure to cyclin dependent kinase inhibitors increases the level of cNICD in the PSM

Having shown that Wnt inhibition or exposure to a panel of CDK inhibitors robustly delays the pace of the segmentation clock, a key question is whether or not there is a common mechanism underlying these observations. We thus investigated whether levels of NICD were perturbed by the CDK inhibitor treatments. Using the same assay as described above, one pool of PSMs was cultured for 3 hr in DMSO and the contralateral halves were cultured in the inhibitor. Both pools were then cultured for an hour in the presence of cycloheximide before Western Blot analysis (*Figure 4C*). Remarkably, and in agreement with model predictions, we observed that the level of cNICD protein was again increased significantly in the presence of Roscovitine (*Figure 4D*, lanes 7, 8; n = 9/10), DRB

**Table 1**. Summary table of densitometry quantifications for cNICD and mNICD in inhibitor assays

| Treatment | N | Mean adjusted relative fold change of cNICD protein | ± standard error of the mean | Statistical test | p-value |
|---|---|---|---|---|---|
| XAV939 | 9 | 3.078 | ±0.776 | One-sample T-test | 0.0280* |
| XAV939 (no CHX) | 5 | 2.491 | ±0.480 | One-sample T-test | 0.0361* |
| Roscovitine | 10 | 1.94 | ±0.255 | One-sample Signed Rank test | 0.002* |
| Roscovitine (no CHX) | 7 | 1.594 | ±0.146 | One-sample T-test | 0.00653* |
| DRB | 8 | 2.557 | ±0.702 | One-sample Signed Rank test | 0.008* |
| DRB (no CHX) | 6 | 1.369 | ±0.081 | One-sample T-test | 0.00601* |
| Cyclopamine | 2 | 1.263 | ±0.106 | One-sample T-test | 0.244 |
| PHA767491 | 2 | 3.479 | ±0.451 | One-sample T-test | 0.115 |
| MLN4924 | 3 | 8.831 | ±2.734 | One-sample T-test | 0.103 |
| Treatment | N | Mean adjusted relative fold change of mNICD protein | ± Standard error of the mean | Statistical test | p-value |
| XAV939 | 8 | 2.221 | ±0.396 | One-sample signed rank test | 0.008* |

*demarcates a statistically significant difference.
Densitometry was performed on Western Blots and the fold change of cNICD or mNICD in the + inhibitor treated sample relative to the corresponding DMSO control was adjusted to the relative change in the tubulin loading control. The given statistical test was used to compare the average value for each assay to a fixed fold-change value = 1 (i.e. no change) and a p-value obtained.

(*Figure 4D*, lanes 9, 10; n = 6/8) and PHA-767491 (*Figure 4D*, lanes 11, 12; n = 2/2) compared to the corresponding DMSO control pool (See *Table 1*). Again, even in the absence of cycloheximide (*Figure 4—figure supplement 1A*), levels of cNICD in the chicken PSM were consistently higher when treated with Roscovitine (*Figure 4—figure supplement 1C*; n = 5/7), and DRB (*Figure 4—figure supplement 1D*; n = 4/6) compared to respective DMSO control lysates (See *Table 1*). Thus, Roscovitine, DRB and PHA-767491 treatments all increase the level of cNICD protein in the chicken PSM, just as they all delay oscillations of the Notch target gene *cLfng*. Hence, taken together the data with the Wnt and CDK inhibitors fulfil the predictions of the model that increased levels of NICD will be associated with an increase to the periodicity of clock gene oscillations.

## XAV939 and the CDK inhibitors extend the clock period in chick leading to formation of fewer somites

To verify the extent to which the period had been extended in the presence of each of the reagents, we performed a time course of the fix and culture assay. We determined the time needed for one full oscillation by extending the time in culture for the cultured half until the expression profiles matched in the two explants, with an extra somite having formed in the cultured explant. Untreated control pairs completed one full oscillation and made an extra somite in 90 min whereas exposure to the reagents resulted in an oscillation period of approximately 120 min (n = 8 for each inhibitor and for DMSO; *Figure 2—figure supplement 3E-J*).

A prediction of the clock and wave-front model of somitogenesis is that a longer clock period will result, assuming a constant wave-front speed, in the formation of fewer, larger somites in a given time. Hence we analysed somite number in drug-treated explants cultured for an extended period of 6 hr and found that fewer somites formed in the presence of Roscovitine (n = 9, mean number = 2.11), DRB (n = 9, mean number = 1.94), PHA-767491 (n = 10, mean number = 2.5) and XAV939 (n = 7, mean number = 3.14) compared to DMSO controls (n = 15, mean number = 4.25). Additionally, using the fix and culture assay to confirm that the clock is still operating under treatment conditions, we observed that *Lfng* expression is still dynamic after the 6 hr culture period (*Figure 5D–H*; n = 5/5 for each

**Figure 5**. Length of newly formed somites are increased following long-term treatment with inhibitors. (**A–C**): Graphs showing log base 2 (fold) changes in 2 consecutive somite length ratios between the average somite lengths for the first (+1), second (+2), and third (+3) formed somites following treatment initiation in the presence of DMSO, XAV939, Roscovitine, PHA767491 or DRB. T-tests were performed to test the null hypothesis that the mean somite length ratio is the same as in DMSO controls. The null hypothesis was rejected in the starred cases ($p < 0.005$) with the following p values XAV = 0.0211; Rosc = 0.0002; Pha = 0.0001; DRB = 0.0007 (**D–H**) Bissected chick caudal explant pairs subjected to the fix and culture assay after 6 hr treatment in the presence of DMSO, XAV939, Roscovitine, PHA767491 or DRB. In each case the cultured side showed *cLfng* expression had advanced. (**I–M**) In situ hybridisation of *cTbx6* on whole embryos following 6 hr treatment in the presence of either DMSO, XAV939, Roscovitine, PHA767491 or DRB. Asterix = newly formed somite(s) during culture, line = length of most recently formed somite, scale bar = 100 µm. (**N–R**) High magnification image of the samples in (**I–M**). XAV939 treated embryos do not form as clearly defined boundaries as the other inhibitor treated embryos. (F) = following cycle.

reagent and for DMSO). Moreover, once the somites from the determined region of the PSM had segmented we observed a significant stepwise increase in somite size along the anteroposterior axis in the embryos cultured in the presence of XAV939, Roscovitine, PHA-767491 or DRB as compared to DMSO control treated embryos (*Figure 5A–C,I–R*). Taken together, these data demonstrate that XAV939, Roscovitine, PHA-767491 and DRB treatments all result in an extension of the clock period and the formation of fewer, larger somites.

## Exposure to XAV939, and the CDK inhibitors increases the half-life of cNICD in the PSM

Given the delays inherent to Notch and Delta-like1 (DLL1) trafficking and their intercellular interaction, it is possible that new NICD could be produced even in the presence of CHX, based on signalling through receptors that were made prior to the initiation of CHX treatment. Thus, if XAV939, Roscovitine or DRB increased the production of DLL1 or Notch1, the observed increase in NICD might be secondary to increased signalling, rather than a result of changes in protein turnover. To address this possibility, we replaced the 1h CHX treatment with a 1h LY411575 treatment to specifically block production of new NICD, repeated the assay and measured levels of cNICD by Western Blot analysis. Thus, one pool of PSMs was cultured for 3 hr in DMSO and the contralateral halves were cultured in XAV939, Roscovitine or DRB. Both pools were then cultured for an hour in the presence of LY411575 before Western Blot analysis (*Figure 4F*). Remarkably, we observed that the level of cNICD protein was again increased in the presence of XAV939 (*Figure 4F*, lane 4), Roscovitine (*Figure 4F*, lane 6) and DRB (*Figure 4F*, lane 8) compared to the corresponding DMSO control pools (*Figure 4F*, lanes 3, 5, 7) where NICD levels were severely diminished due to endogenous NICD degradation plus exposure to LY411575 for an hour. As a control, explants cultured in explant media for 3h followed by 1h LY411575 showed severe depletion of NICD compared to the corresponding DMSO pool of contralateral halves (*Figure 4F*, lanes 1, 2). These data demonstrate that the higher levels of cNICD observed following exposure to the inhibitors is not due to increased production of NICD.

In order to test the hypothesis that the observed increased level of NICD following inhibitor treatment is due to modification of the half-life of NICD, we sought to quantify the half-life of endogenous cNICD in the PSM. Pools of 9 explants from corresponding embryos were prepared as above but cultured in the presence or absence of a given inhibitor. Following an initial 30 min exposure to cycloheximide in both pools to inhibit new protein synthesis, the decay of cNICD protein was monitored as follows: in one pool protein lysate was immediately prepared whilst in the other, culture for an additional 30–60 min was performed before lysate was prepared (*Figure 6—figure supplement 1* and data not shown). Thus, the temporal decay of cNICD was quantified (see *Figure 6A*). Notably the measured profiles are of the form predicted by *Equation (4)*.

From these data we infer the NICD half-life and find that it takes the value 11.87 (+/− 4.97) minutes in untreated control embryos (*Figure 6A*; n = 5). Strikingly, in the DRB, Roscovitine and PHA-767491 treated explants, the half-life of cNICD protein is 29.59 (+/− 9.29), 22.01 (+/− 7.04) min, and 36.41 (+/− 8.88) min, respectively (*Figure 6A,B*; n = 4 for each inhibitor). These data show that DRB, Roscovitine and PHA-767491 treatments all significantly increase the half-life of cNICD protein in the chick PSM. Thus, the data support the model prediction that an increased NICD half-life results in a longer clock period.

## Inhibition of SCF (SKP1- CUL1-F-box protein) E3 ubiquitin ligase complexes results in delayed clock oscillations and higher levels of NICD

Whilst the measurements of NICD half-life are consistent with the model prediction that increased NICD half-life will result in longer period oscillations and increased levels of NICD, we reasoned that if NICD half-life is controlling the clock period, we ought to be able to increase the clock period by targeting NICD turnover.

NICD stability is known to be regulated by the Skp, Cullin, F-box[Sel10/FBXW7] (SCF [Sel10/FBXW7]) E3 ubiquitin ligase complex (*Gupta-Rossi et al., 2001*). We tested whether turnover of NICD within the PSM affects the clock pace by repeating the half-embryo assay in the presence or absence of MLN4924 which inactivates SCF E3 ubiquitin ligase complexes by suppressing neddylation of Cullin-1 (*Soucy et al., 2009*). As a control, we analysed levels of phosphorylated β-catenin which is also targeted for ubiquitination and degradation by the SCF E3 ubiquitin ligase complex. As expected, exposure of PSM explants to 1 μM MLN4924 led to increased levels of phosphorylated β-catenin (*Figure 6D*).

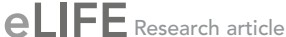

**Figure 6**. cNICD half-life in the PSM is increased after exposure to Roscovitine, DRB and PHA-767491. (**A**): Endogenous cNICD levels analysed by Western Blot (after normalisation to alpha tubulin loading control) in PSM explants cultured initially in the presence of each reagent for 3 hr and subsequently in the presence of inhibitor plus cycloheximide for an additional 30 min before the time-course. Panels show log transformation on repeated data. The half life is proportional to the inverse of the inferred slope (*Equation [4]*). (**B**) Bar chart showing the half-life of cNICD in each condition. Asterix represents statistically significant differences p < 0.05. (**C**) Left Panel: Treatment in the presence (+) or absence (−) of MLN4924 for 3 hr reveals that the inhibitor treated explant has lagging expression of *cLfng* compared to the control contralateral explant. Right panel: After 3 hr in the presence of MLN4924 one PSM explant was fixed while the other was treated for another 45 min, showing *cLfng* expression is still dynamic in the presence of this inhibitor. (**D**) Following the assay described in *Figure 4*, levels of cNICD, normalised to levels of alpha tubulin, in the chick PSM pools are highly increased in the presence of MLN4924. Western Blot analysis of pooled PSM lysates from half PSM explants treated '+' or '−' MLN4924 reveals levels of phosphorylated β-catenin are increased in the chick PSM after 3 hr treatment with MLN4924.

The following figure supplement is available for figure 6:

**Figure supplement 1**. Roscovitine, PHA-767491 and DRB treatment increases the level of cNICD in the chick PSM.

Notably, exposure of PSM explants to 1 µM MLN4924 caused a delay in the pace of *Lfng* dynamic expression across the PSM (*Figure 6C*; n = 16) and increased levels of cNICD (*Figure 6D*; n = 3; *Table 1*). The fix and culture assay confirmed that *cLfng* expression was still oscillatory in the PSM at this concentration (*Figure 6C*; n = 16). Exposure to higher concentrations of MLN4924 (10–100 µM) resulted in progressively greater increases in the level of cNICD but abolished dynamic *Lfng* expression across the PSM (data not shown). Notably, the mathematical model predicts that oscillations are only supported within a limited range of NICD (and Hes7) half-lives (See *Figure 1*). These data support the central thesis provided in this paper: a balance between positive and negative regulators is necessary for oscillations of the somitogenesis clock.

## Reducing NICD production rescues the phase pattern changes and increased level of cNICD in the PSM caused by exposure to XAV939, and the CDK inhibitors

Taken together, the data presented thus far support the hypothesis that increasing the half-life of NICD elevates levels of NICD and lengthens the period of oscillations. In order to directly address the causality between increased NICD stability and the change in pace of oscillations, we attempted to rescue the effect of the inhibitors by modifying NICD production in the PSM.

We performed a titration curve with the gamma secretase inhibitor LY411575 to reduce but not remove NICD production. PSM pools were cultured for 3 hr in DMSO and their contralateral halves cultured in different concentrations of LY411575. In the presence of 5 nM LY411575, cNICD levels decreased significantly compared to the corresponding DMSO pool (*Figure 7A*, lanes 1, 2), while in the presence of 1 nM or 0.1 nM LY411575, the levels of cNICD protein were similar to control DMSO treated contralateral PSMs (*Figure 7B*, lanes 1, 2, *Figure 7C*, lanes 1, 2). Hence, pools of PSMs were cultured for 3 hr in DMSO and the contralateral halves cultured in 10 µM Roscovitine together with 5 nM, 1 nM or 0.1 nM LY411575 prior to Western Blot analysis. The higher levels of NICD usually seen in the presence of Roscovitine alone (*Figure 7D*) are rescued by 1 nM LY411575 such that they resemble the levels of NICD in the corresponding DMSO pool (*Figure 7B* lanes 3, 4). These findings concur with the predictions clearly proposed by our model (*Figure 1H*). In contrast, 10 µM Roscovitine together with 5 nM LY411575 showed severely depleted levels of cNICD compared to the corresponding DMSO pool (*Figure 7A* lanes 3, 4), resembling the effect of 5 nM LY411575 alone (compared to *Figure 7A*, lanes 1, 2). 10uM Roscovitine together with 0.1 nM LY411575 showed increased levels of cNICD compared to the corresponding DMSO pool, resembling the effect of 10 µM Roscovitine alone (*Figure 7C*, lanes 3, 4, compared to *Figure 7D*).

We repeated these experiments and analysed the effect upon the mRNA expression of the clock gene *cLfng*. 5 nM LY411575 severely reduced *cLfng* expression in the PSM (*Figure 8A*; n = 4/4) whereas *cLfng* appeared largely unaffected in the presence of 1 nM or 0.1 nM LY411575, compared to the control explant (*Figure 8B,C*; n = 22/25; n = 11/13, respectively). We then assessed whether we

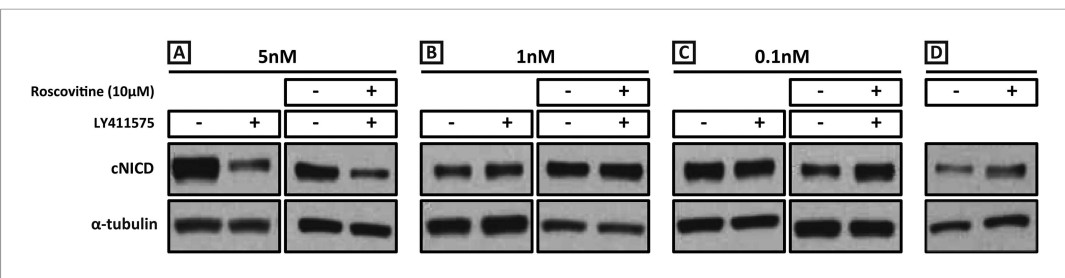

**Figure 7**. Exposure to 1 nM LY411575 rescues the increased levels of cNICD caused by 10 µM Roscovitine. Endogenous cNICD levels analysed by Western Blot (after normalisation to alpha tubulin loading control) in PSM explant pools cultured for 3 hr in the presence (+) or absence (−) of (**A**) 5 nM LY411575 (lane 1,2); 10 µM Roscovitine together with 5 nM LY411575 (lane 3,4) or (**B**) 1 nM LY411575 (lane 1,2); 10 µM Roscovitine together with 1 nM LY411575 (lane 3,4) or (**C**) 0.1 nM LY411575 (lane 1,2); 10 µM Roscovitine together with 0.1 nM LY411575 (lane 3,4) (**D**) or 10 µM Roscovitine alone.

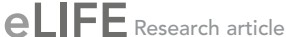

**Figure 8**. Exposure to 1 nM LY411575 rescues the delay in clock gene oscillations caused by Roscovitine, DRB and XAV939. Bissected chick or mouse caudal explant pairs treated '−' or '+' inhibitor and then analysed by insitu hybridisation for *Lfng* mRNA expression: (**A–C**) Treatment of chick PSM explants in the presence (+) or absence (−) of 5 nM LY411575 (**A**), 1 nM LY411575 (**B**), 0.1 nM LY411575 (**C**) for 3 hr reveals that 5 nM LY411575 severely downregulates *cLfng* expression whereas 1 nM and 0.1 nM do not appear to change levels or domain of *cLfng*

*Figure 8. continued on next page*

*Figure 8. Continued*

expression. (**D**–**F**): Treatment of chick PSM explants in the presence (+) or absence (−) of Roscovitine (**D**), DRB (**E**) or XAV939 (**F**) for 3 hr reveals that '+' explants have lagging expression of *cLfng*, with one less somite formed than the '−' explants. (**G**–**I**): Treatment of chick PSM explants in the presence (+) or absence (−) of 1 nM LY411575 together with Roscovitine (**G**), DRB (**H**) or XAV939 (**I**) for 3 hr reveals that 1 nM LY411575 rescues the delay in clock gene oscillations caused by these three inhibitors such that the *cLfng* expression domains in the '−' and '+' explants are very similar. The red arrowheads identify the somites formed during the in vitro culture period of the assay. (P) = previous cycle.

The following figure supplement is available for figure 8:

**Figure supplement 1**. 0.1 nM LY411575 does not rescue the delay in oscillation caused by Roscovitine.

could rescue the changes in phase patterns brought about by Roscovitine, DRB or XAV939 by reducing NICD production through exposure to 1 nM or 0.1 nM LY411575. Strikingly, we observed a rescue of the phase pattern in a significant number of explants cultured in the presence of Roscovitine, DRB or XAV939 together with 1 nM LY411575 (*Figure 8G–I*; n = 15/39, n = 8/20, n = 6/15, respectively) compared to explants cultured in the presence of Roscovitine, DRB or XAV939 alone which show a delay to the phase pattern (*Figure 8D–F*; n = 14/15, n = 9/10, n = 7/8, respectively). Explants cultured in the presence of Roscovitine or XAV939 together with 0.1 nM LY411575 did not rescue the delay caused by these inhibitors (*Figure 8—figure supplement 1* and data not shown; n = 0/9; n = 0/4).

These very striking data showing a rescue of both NICD levels and the delay to the clock oscillations provide strong evidence that increased half-life and higher levels of NICD brought about by Roscovitine, DRB or XAV939 underlie the delay to clock gene oscillations observed on exposure to the inhibitors (*Figure 9*).

## Discussion

Mathematical models that invoke the mechanism of delayed negative feedback have successfully explained numerous experimental observations of the somitogenesis clock (e.g., *Lewis, 2003*; *Monk, 2003*; *Hirata et al., 2004*, *Ay et al., 2013*; *Hanisch et al., 2013*). These models are an excellent example of how a relatively simple mathematical framework can provide insight into the behaviour of a multi-component, multi-scale, nonlinear biological system. An elegant prediction of the models is that the period of the somitogenesis clock can be approximated as a sum of the constituent delays and the half-lives of negative regulators. However, whilst numerous studies have shown that genetic modification of clock components can modify the clock period, relatively little work has been published on whether half-lives of clock proteins have the predicted effect. Moreover, the models neglect the potential role of positive regulators of clock gene expression. In this study we combine experimental manipulation and mathematical modelling to begin to explore these issues.

In order to investigate the role of positive regulators of the somitogenesis clock, we modified previous frameworks to account for the role of NICD in the somitogenesis clock. In the model, NICD and the inhibitory clock protein compete to activate and repress the transcription of a clock gene, respectively. Notably, the release of NICD is a multi-cellular phenomenon involving the periodic and coordinated expression of cell surface proteins (*Bone et al., 2014*). In order to avoid describing each of the constituent components, here we consider a model of a single cell oscillator and assume that NICD is released periodically in a manner that is coupled to the transcription of other clock genes (*Bone et al., 2014*). This formalism allows us to avoid explicitly modelling the temporal dynamics of Delta and Notch across cell populations and focus solely on investigating the role of the half-life of the positive regulator. The predicted outcome of the model is that increasing the half-life of NICD elevates levels of NICD and lengthens the period of oscillations in addition to increasing levels of the inhibitory clock protein. These effects can be rescued by decreasing the production rate of NICD.

In order to explore whether these predictions can be validated experimentally, we have further explored an intriguing phenotype whereby the application of a diverse range of small molecule inhibitors results in a clear, robust delay in the pace of *Lfng* oscillations across the chick and mouse PSM. We find that for each of the small molecule inhibitor treatments, the levels of the positive

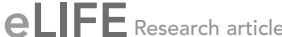

Figure 9. A schematic illustration depicting how modification of NICD stability may cause a delay to the period of the oscillations. (**A**) Schematic showing the competition between molecules of Hes7 and NICD for binding to the promoter of a clock gene. Exposure to Roscovitine, XAV939 or DRB causes increased stability and levels of NICD and thus NICD occupies the promoter for longer than in the wild-type situation, thereby prolonging the oscillation period. Exposure to 1 nM LY411575 together with Roscovitine, XAV939 or DRB reduces the level of NICD production offsetting the increased stability which rescues the delay. t1 = delay due to transcription of Hes7 and Notch; t2 = the

*Figure 9. continued on next page*

*Figure 9. Continued*
delay due to translation of Hes7; t3 = delay due to translation of Notch and subsequent ligand dependent
production of NICD. (**B**) Graphical model depicting how modifying the stability of NICD may cause a delay to the
period of the oscillations by delaying the time-point at which proportional levels of NICD and Hes7 allow switching
of promoter occupancy of these two factors.

regulator, NICD, are elevated and NICD half-life is increased. When interpreted within the framework of the proposed model, these data support the hypothesis that treatment with the small molecule inhibitors results in an increased half-life of the positive regulator NICD, which elevates the levels of NICD, delays the clock, and results in the formation of fewer larger somites. In contrast, when interpreted within the framework of the proposed model, the data do not support the hypothesis, for instance: that the drug treatments solely modify the half-lives of negative regulators of the somitogenesis clock, such as Hes7. We also demonstrate experimentally, in agreement with model predictions, that both the delay in the pace of *Lfng* oscillations across the PSM and the elevated levels of the positive regulator, NICD, resulting from exposure to the small molecule treatments can be rescued by decreasing NICD production.

In this study we observe delays in the PSM phase pattern in the presence of small molecule inhibitors that are rescued by Delta-Notch signalling inhibitors. From these observations we infer that the autonomous frequency of individual oscillations is perturbed as a result of increased NICD stability. Nevertheless, it is also possible that the effect of increasing the NICD half-life is to alter tissue scale properties of the frequency gradient profile (e.g., increasing its rostro-caudal velocity or steepness would result in a greater proportion of cells in the PSM experiencing lower frequencies and thus observations of delayed phase patterns). Regardless of this detail, the premise of this study remains intact: positive as well as negative regulators of the somitogenesis clock can regulate its period.

We note that the inclusion of both positive and negative feedback into the somitogenesis clock model introduces the possibility of interesting nonlinear phenomena that may have relevance to the somitogenesis clock. For instance, Tsai et al. have shown that the inclusion of positive and negative feedback loops can result in the attainment of a wide frequency range over a small range in amplitudes (*Tsai et al., 2008*). This point is potentially relevant in the context of somitogenesis, where a frequency gradient is thought to underlie the striking phase waves that traverse the rostro-caudal axis.

## Wnt inhibition increases levels and half-life of cNICD and extends the period of the clock

Previous reports have suggested that inhibition of Shh signalling slows the segmentation clock in the chick PSM (*Resende et al., 2010*). In this study, we clearly show that, in our hands, inhibiting the Shh pathway does not affect clock pace or the level of cNICD in the chick PSM within the timeframe of our assay. A possible reason for this discrepancy with Resende et al. is that we culture explant pairs for 3 hr whereas they observed delays to somite formation, in the absence of Shh signalling, only after 7.5 hr culture and alterations to clock gene expression after 4.5 hr culture. One possible explanation for the later arising effects they observe on the pace of the clock and somite formation is that they that are indirect, secondary effects arising from Shh inhibition. In contrast, we report Wnt inhibition, within a 3 hr time frame, elicited by exposure to XAV939, increases the half-life of NICD in the chick and mouse PSM, leading to elevated NICD levels and a clear and robust prolonged period of clock gene oscillations and somite formation. Intriguingly, Gonzalez et al. previously reported exposure of mouse PSM explants to LiCl, which acts to inhibit GSK3B, lengthens the period of Hes7 oscillations in the mouse PSM. However, the authors report this effect may rather be via LiCl activation of the MapK pathway which is important for Hes expression in the PSM (*Gonzalez et al., 2013*), since more specific approaches to elevate Wnt signalling have been shown not to change the endogenous period of the segmentation clock (*Aulehla et al., 2008*; *Gibb et al., 2009*). Moreover, Gonzalez et al. found the Wnt inhibitor CKI-7 delayed clock gene oscillations in the mouse PSM, in line with our own observations (*Gibb et al., 2009*; *Gonzalez et al., 2013*).

## CDK inhibitors roscovitine, DRB and PHA-767491 regulate the period of clock gene oscillations

We demonstrate that three CDK inhibitors, Roscovitine, DRB and PHA-767491, also increase the half-life of cNICD and delay the pace of *Lfng* oscillations at concentrations that have no significant effect on cell proliferation. At higher concentrations of inhibitors that did affect proliferation, the oscillations of *Lfng* expression were abolished. Importantly, these data separate the effects of the inhibitors on the segmentation clock from those on the cell cycle. This finding is consistent with the 'clock and wavefront' model of somitogenesis, rather than a cell cycle based model (*Cooke and Zeeman, 1976*; *Stern et al., 1988*).

## Tight control of the half-life of NICD, a positive regulator of oscillatory clock gene expression, appears key to regulating the clock period

A striking finding is the fact that these very different pharmacological reagents share distinctive phenotypic outputs: increased levels and half-life of NICD and extended period of the clock gene oscillations. This raises the compelling prospect that these effects are causally linked. Recent data showing that Notch1 protein is dynamically and periodically expressed in the chick and mouse PSM in a Notch dependent fashion (*Bone et al., 2014*) demonstrate that Notch1/NICD is itself part of a positive feedback loop. We experimentally validate the prediction that under conditions where half-life of NICD is increased, the period of the clock is extended. Our data thus expand upon previous models by highlighting the necessity for an added level of regulation, namely the strict regulation of the half-life for the positive input signal, NICD. Hence, it appears equally important to balance levels of both positive and negative input to correctly regulate the periodicity of clock gene oscillations. This result is similar to observations made in *Nrarp*-null mice, which display increased NICD levels in the PSM and a decrease in somite and vertebral body number although the pace of clock gene oscillations and somite size was not assessed (*Kim et al., 2011*).

Our data with the neddylation inhibitor MLN4924 further substantiate the hypothesis that extended NICD half-life is causally linked to the extended clock period, given that this reagent, which acts via a completely distinct molecular mechanism, nevertheless leads to the same phenotypic output as the Wnt and CDK inhibitors. Thus, in addition to the expected higher levels of NICD in the PSM of MLN4924 treated explants, we saw a corresponding increase in the period of *Lfng* oscillations, as seen for XAV939, Roscovitine, DRB and PHA-767491 treated explants. In addition to elevating levels of NICD, exposure to MLN4924 elevates levels of phosphorylated β-catenin within PSM lysates. Since we show that Wnt inhibition can lead to elevated NICD levels and a delay to clock oscillations, this raises the possibility that the effects of MLN4924 are mediated via Wnt inhibition rather than directly via NICD turnover. However, we propose this is unlikely since in the absence of a degradation pathway the elevated levels of phosphorylated β catenin may in fact continue to participate in target gene transcription. Indeed, APC mutant lines which allow β catenin phosphorylation but are defective in ubiquitination are oncogenic, consistent with the idea that phosphorylated β catenin can indeed activate target genes (*Yang et al., 2006*). Thus, we propose that the extended period of the clock seen following exposure to MLN4924 is due to the elevated levels of NICD brought about via inhibition of NICD degradation. Exposure to higher concentrations of this inhibitor leads to correspondingly higher levels of NICD in PSM explants and this accompanies a loss of dynamic *Lfng* expression. A similar correlation is seen in PSM explants exposed to increasing concentrations of Roscovitine, DRB, PHA-767491. It has previously been shown that overexpression or complete loss of Notch signalling abolishes clock gene oscillations and causes severe segmentation defects (*Niwa et al., 2007*; *Feller et al., 2008*; *Ferjentsik et al., 2009*). These data, together with previous reports investigating the effect of modulating protein levels of key clock components (*Hirata et al., 2004*; *Kim et al., 2011*), indicate that there is a tolerance window for the degree of variance in levels of positive and negative input which will still allow for oscillatory gene expression. These experimental findings are consistent with the predictions of our model which also clearly predict that there is a window within which variations in half-life of both Hes7 and NICD will allow for oscillations. Taken together, our computational and experimental data show that slowing, as opposed to stopping, of the segmentation clock gene oscillations may be brought about through transient and/or partial stabilisation of Notch activity.

The rescue of both phase pattern changes and NICD levels elicited by exposure to both low concentrations of LY411575 together with Roscovitine, DRB or XAV939 provides compelling evidence that increased half-life and higher levels of NICD brought about by these inhibitors underlies the delay to clock gene oscillations (*Figure 9*).

Following exposure to Roscovitine, DRB, PHA-767491 and XAV939, the Western Blot data reveal an additional band for NICD at a slightly lower molecular weight in some of the inhibitor treated samples as compared to the control treated samples. This observation is characteristic of an increase in abundance of a variant of a protein which lacks small post-translational modifications such as phosphate groups. This shift may therefore represent the loss of phosphorylation marks which have been shown to be important for NICD regulation and degradation (*Fryer et al., 2004*; *Jin et al., 2009*). These data support the idea that post translational modification events may regulate the stability of NICD that could then directly affect the pace of the segmentation clock.

## XAV939, Roscovitine, DRB and PHA-767491 may directly affect NICD half-life through post translational modification of NICD

As alluded to above, post translational modifications may underlie the effects of the 4 reagents that each modulate NICD half-life and clock period. We demonstrate that XAV939 treatment elevates levels of GSK3B-mediated phosphorylated β catenin in the PSM and it is possible that it might equally facilitate GSK3B mediated phosphorylation of NICD which has previously been reported to affect NICD stability (*Espinosa et al., 2003*; *Jin et al., 2009*). It is noteworthy that the effect of this phosphorylation event on NICD stability is likely to be context dependent since different groups have reported this modification can increase or decrease NICD stability and it is not known if any of these interactions occur in the PSM (*Foltz et al., 2002*; *Espinosa et al., 2003*; *Jin et al., 2009*). Similarly, it is possible that the CDK inhibitors modulate CDK-mediated phosphorylation of NICD which has also been reported to affect NICD stability (*Fryer et al., 2004*). Interestingly, exposure to specific CDK7, CDK8 or CDK9 inhibitors did not yield the delayed oscillation phenotype or increased levels of NICD and half-life of NICD in the chick PSM (data not shown). These data are somewhat surprising given that there are published reports describing a role for CDK8 in phosphorylating and thereby destabilising NICD. Indeed, CDK8-mediated phosphorylation in the PEST domain of NICD is required to target this protein for ubiquitination and subsequent degradation by the E3 ligase FBW7 (*Fryer et al., 2004*). Whilst it is possible that this mode of regulation of NICD stability by this specific CDK does not occur in the PSM, we acknowledge that it is also possible that the effect on pace and cNICD stability are only made apparent when inhibiting a combination of these activities as seen with Roscovitine, DRB and PHA-76749. It is important to note that an alternative explanation for the lack of effect of specific CDK7, CDK8, or CDK9 inhibitors is that the effect of Roscovitine, DRB and PHA-767491 may be through inhibition of a different set of kinases. The precise identification of the kinase(s) involved in destabilising cNICD will require further investigation.

## XAV939, roscovitine, DRB and PHA-767491 may indirectly affect NICD half-life and the clock period through transcriptional regulation of NICD-stability regulators

It is also formally possible that the common outcome of exposure to these four reagents to influence the pace of the clock involves regulation at the transcriptional level. Roscovitine is known to have highest efficacy against CDK2/CyclinE, CDK5, CDK7/CyclinH and CDK9/CyclinT (*McLue et al., 2002*) whereas DRB and PHA-767491 reduce the kinase activities of CDK7/CyclinH and CDK9/CyclinT towards Serine 2 and Serine 5 on the C-terminal domain (CTD) of RNA pol II. Exposure to DRB, Roscovitine, or PHA-767491 leads to a reduction in Ser5 RNA pol II phosphorylation (*Figure 2—figure supplement 1B*) which is required to promote initiation of transcription as RNA pol II moves away from the gene promoter (*Kormarnitsky et al., 2000*) but it is also associated with 'poised' genes bound by both Ser5 phosphorylated RNA pol II and Polycomb repressor complexes (*Brookes and Pombo, 2012*). Thus, one possibility is that reduced Ser5 phosphorylation of RNA pol II may leave some 'poised' genes in the PSM in a more repressed state. Importantly, certain gene transcripts are more susceptible to DRB treatment than others (*Zandomeni and Weinmann, 1984*; *Chodosh et al., 1989*; *Wada et al., 1998*) and indeed we find not all gene transcripts in the PSM are down-regulated by these inhibitors (e.g., *cLfng* and *cHairy2* expression is delayed in the PSM but not down-regulated).

Thus, the effects of these inhibitors may be to modify certain 'poised' genes to a more repressed state that specifically include genes encoding factors required for destabilising NICD.

Given that XAV939 elicits the same phenotypic output as exposure to the CDK inhibitors it is conceivable that all four reagents affect the transcription of some common genes; some involved in regulating the clock while others not; indeed we identified one gene commonly affected by all 4 reagents namely *Gli1* (known to be regulated by Wnt signalling in the neural tissue and somites (*Munsterberg et al., 1995*), *Figure 2—figure supplement 1E–G*). For this particular gene we show it is unlikely to be involved in clock regulation (*Figure 3*). However, a transcriptomic analysis of common target genes would allow this possibility to be fully explored to identify genes co-regulated by these reagents which potentially encode regulators of NICD stability in the PSM.

Finally, a recent report has beautifully demonstrated that in zebrafish embryos the rate of tissue shortening, in addition to the periodicity of clock gene oscillations, is important in determining the periodicity of segmentation (*Soroldoni et al., 2014*). Further analyses are required to investigate if this is a conserved feature in all vertebrate species. That analysis notwithstanding, our data nevertheless clearly show that the pace of the segmentation clock itself has a strong influence upon the pace of somite formation since significantly fewer somites are formed within a given time-frame when the pace of clock gene oscillations and NICD stability are affected. In conclusion, our data suggest that the stability and turnover of NICD is inextricably linked to the regulation of the pace of clock gene expression across the PSM.

## Materials and methods

### Experimental Procedures

#### Chick culture

Fertilized chick (*Gallus* gallus) embryos from Winter Farm, Cambridge, UK, were incubated for approximately 40 hr at 38℃ 5% $CO_2$ to yield embryos between Hamburger Hamilton (HH) stages 10–12 as judged by somite number and morphology. Explants were prepared as previously described (*Palmeirim et al., 1997*). In brief, for the half-embryo assay: the caudal end of the chicken embryo is isolated to contain the most caudal somite pairs and the tail tissues. The isolated tail is then bisected down the midline along the neural tube to produce two identical left and right half-PSM explants which allows an internal control from the same embryo with which to compare the oscillation phase of *cLfng* expression by in situ hybridisation, following culturing of the two halves in various culture conditions. The following small molecule inhibitors were typically used at the given concentrations and dissolved in Dimethyl sulfoxide (DMSO) unless otherwise stated: 150 nM LY411575 unless otherwise stated (generated in house, University of Dundee; *Ferjentsik et al., 2009*; *Bone et al., 2014*), 1 µM MLN4924 (generated in house, University of Dundee); 20 µM Cycloheximide (Tocris Bioscience; dissolved in ethanol); 100 µM XAV939 (Tocris Bioscience); 25–50 µM Cyclopamine (Calbiochem); 10 µM Roscovitine (Calbiochem); 10 µM 5,6-Dichloro-1-beta-D-ribofuranosylbenzimidazole (DRB) (Sigma, United Kingdom); 1 µM PHA-767491 (kind gift from Nathanael Gray, HARVARD.EDU). Titration assays were performed to establish the lowest concentration that abolishes expression of control target genes. For Fix and culture: both half-PSM explants were treated with inhibitor but while one was removed from culture after 3 hr ('fix'), the other side was cultured for an additional 45 min ('culture') which corresponds to half of the usual 90 min cycle for a complete oscillation of *cLfng* in the chicken PSM. Variations include culturing the 'culture' side for '90' or '120' min to determine the length of an oscillation. For the Western Blot analysis half-PSMs from 9 embryos were cultured in DMSO control vehicle and the corresponding 9 half-PSMs cultured in the presence of a small molecule inhibitor. Following 3 hr of culture, both pools were treated with 20 µM Cycloheximide for an additional 30 min before protein lysates were prepared simultaneously. For the half-life assay: half-PSMs from 9 embryos were pooled and both pools were cultured in each of the reagents for 3 hr and following 30 min exposure to cycloheximide, the decay of cNICD protein was monitored in the presence of these inhibitors by lysing PSM explants at different time points. Experiments were conducted in strict adherence to the Animals (Scientific Procedures) Act of 1986 and UK Home Office Codes of Practice for use of animals in scientific procedures.

#### In situ hybridisation

Whole-mount in situ hybridisations utilising exonic and intronic probes were performed as described (*Gibb et al., 2009*).

## Phospho-HistoneH3 staining

Fixed PSM explant tissue was proteinase K (Roche) treated and fixed (4% formaldehyde in PBS, 2 mM EGTA, 0.1% glutaraldehyde [Sigma]) prior to washing in PBST. Explants were then blocked in 2% bovine serum albumin (BSA) in PBST for 2 hr at room temperature. Anti-phospho histone-H3 antibody (Upstate) was then added at 10 µg/ml and the samples incubated at 4°C overnight. Specimens were then washed for 10 hr in a minimum of three changes of PBST before the Alexa-fluor488 conjugated mouse anti-rabbit antibody (Invitrogen) was added at 2 µg/ml in PBST and left overnight at 4°C. Samples were then washed for 10 hr in PBST with a minimum of 3 changes of solution. Explants were then mounted on SuperFrost microscope slides (VWR) using Hydromount (National Diagnostics) and imaged using either a compound fluorescence microscope (Leica DM5000 B) or a Zeiss 710 confocal microscope. Images were quantified using Volocity v6.0.1 imaging software (Perkin Elmer) and either t-tests or Mann–Whitney tests performed using SigmaPlot v12.0 software.

## NucView apoptosis staining

+/− treated explant pairs were cultured for 2 hr (chicken) or 3 hr (mouse) before the addition of 5 µM NucView 488 Caspase-3 Assay Kit for live cells (Biotium) and cultured for an additional 1 hr. Samples were then fixed for 1 hr at 4°C in 4% formaldehyde/PBS. Samples were then washed into PBST and explant pairs were mounted on superfrost slides (VWR) in Prolong Gold anitfade reagent (Molecular Probes) for imaging on a Zeiss 710 confocal microscope. Quantification of NucView cell staining was performed using Volocity v6.0.1 imaging software (Perkin Elmer) and statistical analyses done using either t-tests or Mann–Whitney tests with Sigmaplot v12.0 software.

## Generation of cNICD antibodies

The peptide VLVSRKRRREHGC (corresponding to residues 1792–1803 of the chicken NOTCH1 protein, with the addition of a C-terminal cysteine residue for coupling) was coupled separately with Keyhole Limpet Hemocyanin (KLH) and BSA using m-maleimidobenzoyl-N-hydroxysuccinimide ester (MBS) and dialysed into PBS. The affinity purification of the cNICD antibody was achieved by putting the third bleed serum from the rabbit which had been immunised with the peptide on an affinity column which had been previously prepared by coupling the cNICD peptide with vinylsulfone activated sepharose. The purified antibody was then dialysed into PBS.

## Western Blot analysis

Pooled samples of 9 Chick embryo half-PSM explants were prepared by pipetting treated tissue in 90 µl mixture containing lysis buffer (50 mM Tris Hydrochloride pH7.5, 150 mM Sodium chloride [NaCl], 1 mM ethylenediaminetetraacetic acid [EDTA], 1 mM ethylene glycol tetraacetic acid [EGTA], 10 mM Sodium pyrophosphate, 10 mM Sodium glycerophosphate, 50 mM Sodium fluoride, 1 mM Sodium orthovanadate, 0.3% 3-[{3-cholamidopropyl} dimethylammonio]-1-propanesulfonate [CHAPS], 1 mM Benzamidine, 0.1% β-mercaptoEtOH, 0.1 mM phenylmethanesulfonylfluoride [PMSF]), supplemented with 1x Sample Reducing Agent and 1x LDS Sample Buffer (Life Technologies). Samples were then reduced at 95°C for 5 min and allowed to cool briefly before loading 20 µl of each sample onto 4–12% Bis–Tris acrylamide gels (Life Technologies). Gels were then Blotted using standard molecular techniques and the resultant nitrocellulose membranes (Whatman) were blocked in 5% milk in TBS–0.1% Tween 20 (TBST). Membranes were treated overnight at 4°C in one of the following antibodies diluted in 5% BSA in TBST: 1:1000 rabbit anti-chick NICD (University of Dundee); 1:1000 rabbit anti-mouse NICD (Cell Signalling); 1:1000 rabbit anti-phospho S33, S37, T41 β-catenin (Cell Signalling); 1:5000 rabbit anti-total β-catenin (I. Nathke); 1:1000 rabbit-anti Fbw7 (Abcam); 1:1000 rabbit anti-mouse Hes7 (University of Dundee, generated as *Bessho et al., 2003*); 1:200 rat-anti phospho S2 RNA pol II (Chromotek); 1:200 rat-anti phospho S5 RNA pol II (Chromotek); 1:200 rat-anti total RNA pol II (Chromotek); 1:5000 mouse anti-α-tubulin (Abcam). Membranes were then incubated with the appropriate secondary antibody (HRP) in 5% milk in TBST and standard ECL revelation (Pierce).

## Statistical approach

It was hypothesized that half-life of NICD would alter as a function of the treatments applied. Small samples and short time series limited the ability to develop a robust empirical model that could be broadly applied; as such each half-life was developed on a per treatment basis with linear modelling. Natural log transformations were performed on concentrations such that a slope and intercept could be reasonably estimated. Those with significant slopes are reported in *Figure 6*. Statistical analyses were conducted in R 3.2.0 (*R Core Team, 2013*).

## Mathematical model

We develop a minimal model of the somitogenesis clock with the goal of exploring the roles of both negative and positive feedback of clock products on clock dynamics. The model has two dependent variables: numbers of cytoplasmic clock protein, $p(t)$, and NICD, $n(t)$. Given that cytoplasmic clock protein and NICD act as a transcriptional repressor and activator, respectively, we assume that the rate of production of clock gene products is proportional to

$$f(p, n) = \frac{\left(\frac{n}{K_n}\right)}{1 + \left(\frac{p}{K_p}\right)^2} , \tag{1}$$

where $K_p$ is an IC50 constant and $K_n$ is a constant that determines a scale for NICD activation. We note that the competition between activation and repression of transcription by NICD and clock protein, respectively, that is encapsulated by *Equation (1)* plays a key role in the model behaviour.

Assuming linear decay of cytoplasmic protein at rate $k_2$, the rate of change of numbers of cytoplasmic clock protein with respect to time is given by the delay differential equation

$$\frac{dp}{dt} = k_1 f(p(t - T_1), n(t - T_1)) - k_2 p(t), \tag{2}$$

where the delay $T_1$ represents the time that elapses between protein/NICD influencing transcriptional activation and the effect on cytoplasmic protein levels (i.e. the sum of transcription, splicing, export and translation delays).

Whilst it is well established that NICD undergoes pulsatile dynamics in the PSM, Notch1 and Dll1 undergo qualitatively similar behaviour and Notch is a target of Notch signalling (*Bone et al., 2014*). The production of NICD within a given cell is therefore a multicellular process with multiple oscillatory regulatory inputs. We capture the periodic production of NICD by making the simplifying assumption that the production rate of NICD is the same as that of clock protein but with a longer delay, $T_2$, to account for the trafficking of Notch1 and Dll1 to the membrane and their transactivation. Assuming that NICD decays at rate $k_4$, we obtain

$$\frac{dn}{dt} = k_3 f(p(t - T_2), n(t - T_2)) - k_4 n(t). \tag{3}$$

We use this model (*Equations (1)-(3)*) to investigate how the decay kinetics of both positive and negative regulators of clock transcription can influence the oscillator period. Further model details can be found in the *Supplementary file 1*.

## Measuring the half-life of endogeneous NICD

In order to measure the half-life of endogeneous NICD, the following experimental protocol is followed: 9 embryos are dissected along their midlines with contralateral halves placed in two pools (A and B). The pools are initially cultured for 3 hr, either in the presence of a drug or control medium, and then for 30 additional minutes in the presence of a translation inhibitor. The samples from pool A are homogenised and the total amount of NICD in a given pool is quantified. Pool B is then cultured for a further time $\tau$ (30, 40 or 60 min), homogenised and NICD is quantified.

Consider Pool A and let $A_i(t)$ represent the amount of NICD in the $i^{\text{th}}$ PSM at time $t$ (where $t=0$ represents some arbitrarily chosen position in the periodic NICD cycle). The (experimentally measurable) total amount of NICD in the whole of Pool A is given by

$$\overline{A} = \sum_i^N A_i(t),$$

where $N$ is the number of samples in the pool. Similarly, the total amount of NICD in Pool B is given by

$$\overline{B} = \sum_i^N B_i(t).$$

As the samples in Pools A and B are paired (contralateral halves), $B_i(t)=A_i(t+\tau)$, where $\tau$ is the additional culture time for Pool B, hence

$$\overline{B}(\tau) = \sum_i^N A_i(t+\tau).$$

Calculating the ratio of total amount of NICD in Pool B to that of Pool A, we define

$$F(\tau) = \frac{\overline{B}(\tau)}{\overline{A}}.$$

As each of the nine tissue samples can be thought of as being random samples from some underlying periodic NICD time series, $A(t)$, the above sums can be thought of as representing a naive Monte Carlo integration. Hence the experimentally calculated $F(\tau)$ is an approximation to

$$F(\tau) = \frac{\int_0^T A(t+\tau)\,dt}{\int_0^T A(t)\,dt},$$

where $T$ represents the period of the oscillations. The question now is whether the half-life of NICD can be inferred by experimentally measuring $F(\tau)$ at different values of $\tau$.

In the case where $A(t)$ is periodic, both integrals will tend to the same value and the measure tends to one. In contrast, if production of NICD ceases, and NICD undergoes only linear decay, $A(t+\tau)=A(t)e^{-\beta\tau}$, where $\beta$ is the NICD degradation constant. In this case,

$$F(\tau) = \frac{\int_0^T A(t)e^{-\beta\tau}\,dt}{\int_0^T A(t)\,dt} = e^{-\beta\tau},$$

and the measure decays exponentially. Combining, if production stops at some time $t_s$ after the experiment begins, we obtain the measure

$$F(\tau) = \begin{cases} 1, & \tau < t_s \\ e^{-\beta(\tau-t_s)}, & \tau \geq t_s. \end{cases}$$

Taking natural logs we obtain

$$\ln(F(\tau)) = \begin{cases} 0, & \tau < t_s \\ -\beta(\tau - t_s), & \tau \geq t_s. \end{cases} \tag{4}$$

Hence linear regression allows the determination of the decay constant $\beta$ and the corresponding half-life is calculated to be $\ln 2/\beta$.

Whilst the above equation predicts the experimental measure in the case of a large number of samples, it is notable that the effective finite sampling of the integrals could result in large measured deviations from mean behaviour. In fact, noting that NICD dynamics are strongly pulsatile, the above integrals will likely be dominated by the number of sampled NICD peaks in a given pool.

## Acknowledgements

We are grateful to the JKD laboratory. Special thanks also go to David Booth for assistance with statistical analyses, to Lucas Morales Moya, Rich Green and Jamie Carrington for experimental assistance, to K Storey, C Weijer, A Muller and M Maroto for critical reading of the manuscript. We thank the groups of N Gray, K Storey, O Pourquie, D Ish Horowicz, D Alessi for reagents, Marianne Reilly and Nikoletta Patourgia for their administrative support. We thank Pedro Miguel Branco Barbacena for optimising the cNICD antibody. This work was supported by College of Life Science alumnus studentship/ a BBSRC studentship to GW; MRC studentship to RB; a WT project grant to JKD WTssss57MA. The work was also supported by a Welcome Trust Strategic award **097945/Z/11/Z**.

## Additional information

### Funding

| Funder | Grant reference | Author |
|---|---|---|
| Wellcome Trust | 097945/Z/11/Z | J Kim Dale |
| Wellcome Trust | WT089357MA | J Kim Dale |
| Medical Research Council (MRC) | studentship | Robert Alexander Bone |
| Biotechnology and Biological Sciences Research Council (BBSRC) | studentship | Guy Wiedermann |

The funders had no role in study design, data collection and interpretation, or the decision to submit the work for publication.

### Author contributions

GW, Conception and design, Acquisition of data, Analysis and interpretation of data; RAB, JCS, Acquisition of data, Analysis and interpretation of data, Drafting or revising the article; MB, Technician for the group and Key member of the team in all projects, Acquisition of data; PJM, JKD, Conception and design, Analysis and interpretation of data, Drafting or revising the article

### Author ORCIDs

J Kim Dale, http://orcid.org/0000-0002-9294-947X

### Ethics

Animal experimentation: This study was performed under a home office license issued to Dr Kim Dale and under the regulations of the home office Animal Welfare ACT. All of the animals were handled according to approved institutional animal care and use committee (IACUC) protocols of the University of Dundee. The protocol was approved by the ethical committee of the University of Dundee. Every effort was made to minimize suffering.

## Additional files

### Supplementary file

• Supplementary file 1. This file contains further analysis of the mathematical model and notes on parameter identification.

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
