## [Decision Letter]

Thank you for sending your work entitled “A balance of positive and negative regulators determines the pace of the segmentation clock” for consideration at *eLife*. Your article has been evaluated by Janet Rossant (Senior Editor) and three reviewers.

The reviewers all agreed that there was considerable merit in your study and that the concept of a positive regulator of the somite clock was an interesting one. However, there was one major concern. The experimental work demonstrates nicely a correlation between NICD levels and stability and period length. Nonetheless, all reviewers were concerned that you had not really proven a causal relationship between elevated NICD half-life and extension of the oscillation period.

There was a quite extensive discussion on this issue, and the reviewers were not sure exactly how you could prove your hypothesis. Some ideas were introducing a low level of RNAi to Notch1 to reduce Notch levels or treating with a gamma-secretase inhibitor to reduce NICD production or to increase FBXW7 levels. Alternately, another hypothesis discussed was to show that the clock period can be returned to normal by making NICD less stable (despite treating with a CDK or Tankyrase inhibitor). Other possibilities raised were locally raising NICD levels or changing half-life by removing the PEST domain. We realize that these are not necessarily the right experiments, and therefore would like to give you the option to respond to this criticism. Please take into account the major concerns raised by reviewers 1 and 2, appended below.

Reviewer 1:

1) The percentage of embryos where *cLfng* expression is delayed after (for example) Wnt inhibition is almost certainly statistically significant, but in the absence of data from control embryos (both halves cultured in DMSO) and some statistical comparison, this cannot be confirmed. The authors should already have this data.

2) For figures looking at NICD levels and half-life following drug treatment, it is not entirely clear to me that treatment with cycloheximide alone will prevent production of new NICD in explants. Given the delays inherent in Notch and DLL1 trafficking, cell surface presentation, and interactions, it seems possible that new NICD could be produced even in the presence of CHX, based on signaling through receptors that were made prior to initiation of CHX treatment. Thus, if any of the inhibitors increased the production of DLL1 or NOTCH1, for example, then the increases in NICD observed might be secondary to increased signaling, rather than a result of changes in protein turnover. Control experiments where inhibition of Notch signaling was initiated at the time of CHX treatment could address this concern, and provide additional support to the thesis that the effects the authors observe are due to changes in NICD half-life.

3) The use of *p* values in Figure 4 legend indicates that multiple pools were assessed and the Western Blot bands were quantified. The details of this process, as well as the averages and standard deviations of the values and the statistical analysis used should be described. Further, I don't find the phosphoserine 5 controls showing activity of CDK inhibitors in Figure 2—figure supplement 1 extremely convincing. Details of quantification of Western Blots might help here as well. The authors should have this data.

4) From the images provided in Figure 5, it is not clear how “normal” the longer somites formed in the presence of various inhibitors actually are. Given that changes in clock synchrony also can affect somite size and morphology, the reader needs to have a clear idea of how (if) somite production might be perturbed by these treatments. Images that focus more closely on the region of interest to let the reader interpret morphology, and perhaps in situ analysis with markers that would delineate somite compartments would lend support to the authors’ conclusions.

5) Data showing that MLN4924 increases phospho-β catenin are shown in Figure 6, but as far as I could tell, are not discussed? Since the authors claim that Wnt inhibition can on its own increase NICD levels and half-lives, this data should be at least acknowledged and put in some kind of context.

Reviewer 2:

1) The experimental work nicely shows a correlation between NICD levels and stability and period length. It is however not clear if there is a causal relationship between elevated NICD half-life and extension of the oscillation period. The last set of experiments presented (subsection “Inhibition of SCF (SKP1- CUL1-F-box protein) E3 ubiquitin ligase complexes results in delayed clock oscillations and higher levels of NICD”), aiming at inhibiting the SCF E3 ubiquitin ligase complex is extremely important to address this issue. However, it should be extended to show that the inhibitor does not work through inhibition of the Wnt pathway, as is the case of the previous inhibitors. Otherwise the order of events remains unclear. Related to this, the last panel of Figure 6, which is not described in the Results section (and should be), suggests that levels of phosphorylated β catenin are elevated following MLN4924 treatment. This should be explained.

2) In many ways, Figure 1 is essential to this manuscript. We feel it could be enhanced to facilitate the reading of the manuscript and understanding of the model and predictions. Figure 1 are not visually explicit enough to understand: a) the relationship between the clock protein, NICD, the feedback loops and the output on the oscillation period; b) how this model differs from the previous ones (Figure A could be replaced by the previous models); c) what the assumptions underlying this new model are. In addition, and maybe in the Experimental Procedures section, a statement explaining how the model can fit both the chicken and mouse somitogenesis, which oscillations for instance have a different period, would be useful.

3) Figure 1 should be explained in a bit more detail, since this is the basis for understanding the rationale behind the experiments. The period on the top of the plots C-E correspond to what? In D and E how much are the half-lives of NICD and Hes7 increased? Maybe it should be stated clearly what the used values are and that the loss of oscillations in Figure 1 corresponds to red region in Figure 1 near the solid line that is the limit of oscillations.

4) The competition relationship between NICD and the inhibitory clock protein to activate and repress transcription of a clock gene respectively (what parameter/properties considered?) impinges on the predictions of the model and interpretation of Figure 1, which postulates that elevated NICD half-life results in more clock protein, and should be better spelled out. In addition, depending on the ability of the clock protein to be a potent transcription inhibitor, having NICD longer around but not much more NICD protein may not have much of an impact. It would be useful to explicit which parameter(s) in the mathematical model reflect this aspect.

5) The description of the approach to measure the half-life is a bit confusing and hard to follow with regard to the equations. *A*_*i*_*(t)* is the amount of NICD in both pools A and B. “t” is a free parameter. We suggest for the second equation in the subsection “Measuring the half-life of endogeneous NICD”: B_bar(*τ*)=A_bar(t+τ)=sum… and F(τ)=A_bar(t+τ)/A_bar(τ). This is more coherent with the integral form of F(τ) in the fourth equation.

Maybe it would be useful to have a profile sketch for A as a function of t. If we understand correctly, as long as τ<*t*_*s*_, A is equal to a constant value A_0_, then as we reach to *t*_*s*_, A(*t*) decays exponentially. Using this sketch, it will be easier to understand that for τ<=*t*_*s*_, A(t) = A(t+τ), else A(t+τ)=A(t)e^–^β^τ^). We also suggest that the first sentence of the second paragraph changes to “Letting *A*_*i*_*(t)* be the amount of NICD in the i^th^ tissue sample at time (*t*). We define the experimentally measurable sum in pool A as…”

6) What is T in the integral form of F(τ) in the subsection “Measuring the half-life of endogeneous NICD”, the clock period?

7) Figure 6: It should be mentioned that the vertical axis is log F(τ), the units of the horizontal axis is minutes. The label seems to be “τ” and not “t”. It would be useful to add at the end of the legend (A) that the inverse of the slope gives the half-life (data presented in Figure 6). I don't understand where does the 0.5C_0_=C_0_e-^slope t^ relationship come from. In the main text, Log F(τ)=–β(Τ–*t*_*s*_). From the slopes of Figure 6 we get the values in Figure 6 and from the intercept we should be able to get *t*_*s*_ (which seems to be of the order of 10 minutes). Can the authors comment on this (and why this has not been used to calculate *t*_*s*_?).

8) Figure 4 and Figure 6 seem to point to different values for NICD half-life. The 2nd lane in Figure 4 indicates that all NICD proteins are cleared in 3 hours; the 4th lane indicates that significant residual NICD remains 1h after its production has been blocked. Could the authors explicit how these experimental results fit with the calculated half-life indicated in the subsection “Exposure to XAV939, and the CDK inhibitors increases the half-life of cNICD in the PSM”?

In addition, it would be very informative to visualize (in embryos) if the NICD half-life is homogenously increased in the somites and presomitic mesoderm, as its impact on the oscillation period is expected in a very defined spatial window.

9) Impact of SHH inhibition on the oscillation. It is a bit unsettling to only find in the Discussion a reference to the 2010 PNAS paper claiming the opposite results, and not upfront in the Results section. In addition, one would have liked to read possible reasons for the discrepancy.

---

## [Author Response]

*The reviewers all agreed that there was considerable merit in your study and that the concept of a positive regulator of the somite clock was an interesting one. However, there was one major concern. The experimental work demonstrates nicely a correlation between NICD levels and stability and period length. Nonetheless, all reviewers were concerned that you had not really proven a causal relationship between elevated NICD half-life and extension of the oscillation period*.

There was a quite extensive discussion on this issue, and the reviewers were not sure exactly how you could prove your hypothesis. Some ideas were introducing a low level of RNAi to Notch1 to reduce Notch levels or treating with a gamma-secretase inhibitor to reduce NICD production or to increase FBXW7 levels. Alternately, another hypothesis discussed was to show that the clock period can be returned to normal by making NICD less stable (despite treating with a CDK or Tankyrase inhibitor). Other possibilities raised were locally raising NICD levels or changing half-life by removing the PEST domain. We realize that these are not necessarily the right experiments, and therefore would like to give you the option to respond to this criticism.

We have given the referees’ concerns careful consideration and conducted a key experiment to address the major issue that has been raised (i.e. causality of increased NICD stability and change in the pace of the clock – please see below for details). In short we titrated the concentration of the gamma secretase inhibitor LY411575 so as to reduce the elevated levels of NICD that were induced by the Rosc/DRB/XAV treatment and found that the clock delay phenotype is reversible. We believe that this dose-dependent rescue using a drug that is highly specific to NICD production in the cytoplasm provides strong evidence in support of the main thesis of our study that the increased NICD stability induced by Rosc/DRB/XAV treatment leads to a change in the pace of the clock. Moreover, the results from the rescue are consistent with the predictions of the mathematical model (see Figure 1).

In addition, we have responded to all other comments, added 4 new figures, a table and new data to Figures 4 and 5. Moreover, we have refined the mathematical model so that it has fewer variable and parameters and can better illustrate the major thesis of this study.

A key issue raised is that of causality between NICD stability and the change in pace of oscillations. We have used a small molecule approach to rescue the phenotype as follows:

LY411575 is a specific gamma-secretase inhibitor that blocks the release of NICD into the cytoplasm. We titrated its concentration so as to reduce but not remove NICD production in an attempt to offset the stability caused by Roscovitine/DRB/XAV939. The aim was to attempt to rescue the delay induced by the Roscovitine/DRB/XAV939 treatment.

In the initial draft of the manuscript we showed that Roscovitine, DRB or XAV939 delay the clock (approximately 95% chick; 90% mouse). In this revised draft we show that this phenotype is rescued in a significant proportion of the explants (30–40%) by the addition of 1 nM LY411575. Higher concentrations of LY411575 together with Roscovitine result in severe downregulation of *Lfng* mRNA expression whilst lower concentrations of LY411575 (0.1 nM) do not rescue the delay effect of Roscovitine (Figure 8 and Figure 8—figure supplement 1). Thus we have identified a dose-dependent rescue using an inhibitor that is specific to NICD production within the cytoplasm.

Additionally, we have run Western Blots to show that the higher levels of NICD, usually seen in the presence of Roscovitine alone, are rescued by adding in 1 nM LY411575 (Figure 7). Given the well-established role of LY411575 in preventing NICD cleavage, we argue that this rescue is a particularly striking result that provides extremely strong support for the central thesis proposed in this study: that increased half-life and higher levels of NICD brought about by Roscovitine, DRB or XAV939 underlie the delay to clock gene oscillations observed on exposure to these inhibitors.

Furthermore, the result of this rescue is in agreement with the prediction of the mathematical model (see Figure 1) which suggests that decreasing production of NICD in a background of longer half-life NICD will result in rescue of the phenotype (i.e. it is the ratio of production to decay rates that is the important parameter, hence reducing the decay rate can be offset by reducing the production rate).

We note that in order to demonstrate causality, the reviewers’ suggested the following experiments:

i) Introduce a low level of RNAi to Notch1 to reduce Notch levels;

ii) Treat with gamma-secretase inhibitor to reduce NICD production;

iii) Increase FBXW7 levels;

iv) Show the clock period can be returned to normal by making NICD less stable despite treating with a CDK or tankyrase inhibitor;

v) Locally raise NICD levels; and

vi) Change half-life by removing the PEST domain.

We gave each of these suggestions careful consideration. However, a technical problem with a number of the suggestions is that they require electroporation of a construct into the PSM. Despite huge experience working on this system in our lab, we find that this technique targets cells in a mosaic fashion and is at best 50% efficient. In such an experiment we would have no control over the amount of construct taken up.  Moreover, some of the suggested constructs would then need to compete with the endogenous Notch protein.

In our experience with this system, an effective way to affect all cells in a temporally controlled manner is through the use of small molecule inhibitors. In the future we would of course aim to identify the mechanism by which NICD is stabilised and generate a mouse that phenocopies this mechanistically in order to conclusively demonstrate causality. However, that is out of the remit of this manuscript.

Reviewer 1:

*1) The percentage of embryos where* cLfng *expression is delayed after (for example) Wnt inhibition is almost certainly statistically significant, but in the absence of data from control embryos (both halves cultured in DMSO) and some statistical comparison, this cannot be confirmed. The authors should already have this data*.

We have included these data. In DMSO treated explant pairs an asymmetric pattern of *cLfng* expression was rarely seen (n = 2/18) showing asymmetry, data not shown) and as such the effects of XAV939 on *cLfng* expression could not be attributed to DMSO or natural variability in expression or to the assay itself.

*2) For figures looking at NICD levels and half-life following drug treatment, it is not entirely clear to me that treatment with cycloheximide alone will prevent production of new NICD in explants. Given the delays inherent in Notch and DLL1 trafficking, cell surface presentation, and interactions, it seems possible that new NICD could be produced even in the presence of CHX, based on signaling through receptors that were made prior to initiation of CHX treatment. Thus, if any of the inhibitors increased the production of DLL1 or NOTCH1, for example, then the increases in NICD observed might be secondary to increased signaling, rather than a result of changes in protein turnover. Control experiments where inhibition of Notch signaling was initiated at the time of CHX treatment could address this concern, and provide additional support to the thesis that the effects the authors observe are due to changes in NICD half-life*.

We appreciate this comment and have conducted this experiment. We have replaced the 1h CHX treatment with 1h LY411575 treatment following exposure to Roscovitine, DRB or XAV939 and we then conducted the western to look at levels of NICD. We observe a very similar result to that seen with CHX: In contrast to explants cultured in explant media for 3h followed by 1h LY411575 which show severe depletion of NICD, we see that explants cultured in Roscovitine, DRB or XAV939 for 3h followed by 1h LY411575 show remarkably higher levels of NICD (see Figure 4). This data together with the half-life assay demonstrates that the higher levels are due to increased stability rather than increased production of NICD.

*3) The use of* p *values in*
Figure 4
*legend indicates that multiple pools were assessed and the Western Blot bands were quantified. The details of this process, as well as the averages and standard deviations of the values and the statistical analysis used should be described. Further, I don't find the phosphoserine 5 controls showing activity of CDK inhibitors in*
Figure 2—figure supplement 1
*extremely convincing. Details of quantification of Western Blots might help here as well. The authors should have this data*.

We have included a table with these details (please see Table 1).

*4) From the images provided in*
Figure 5*, it is not clear how “normal” the longer somites formed in the presence of various inhibitors actually are. Given that changes in clock synchrony also can affect somite size and morphology, the reader needs to have a clear idea of how (if) somite production might be perturbed by these treatments. Images that focus more closely on the region of interest to let the reader interpret morphology, and perhaps* in situ *analysis with markers that would delineate somite compartments would lend support to the authors’ conclusions*.

We have added new panels to Figure 5 in which we present higher magnification images of the somites formed during the culture period.

*5) Data showing that MLN4924 increases phospho-β catenin are shown in*
Figure 6*, but as far as I could tell, are not discussed? Since the authors claim that Wnt inhibition can on its own increase NICD levels and half-lives, this data should be at least acknowledged and put in some kind of context*.

We have reported the effect of MLN4924 upon phospho-β catenin in both the Results and Discussion. Phosphorylated β catenin is targeted for ubiquitination and degradation by the E3 ligase complex which is targeted by MLN4924. Thus, it is expected that exposure to this inhibitor leads to increased levels of phospho- β catenin. However, we propose the effects of MLN4924 on levels of NICD and disruption of phase patterns are not via indirect effects upon Wnt signaling for the following reasons: elevated levels of phospho βcatenin are indicative of Βcatenin molecules that would normally be decommissioned from participating in Wnt signaling but which are prevented from being degraded by MLN4924. Moreover, elevated levels of phospho βcatenin could in principle continue to participate in target gene transcription in the absence of a degradation pathway. If so it would mean MLN4924 potentially leads to elevated levels of Wnt signalling. Indeed, there are some APC mutant lines which allow βcatenin phosphorylation but are defective in ubiquitination ([68] JBC). Given that these mutations are oncogenic, it would seem that phosphorylated βcatenin can indeed activate target genes. Even if phospho βcatenin cannot participate in target gene transcription, the phosphorylation decommissioning event is independent of MLN4924 and is thus happening at physiological levels and so again is not indicative of an inhibitor mediated block to Wnt signalling. Thus, we propose it is unlikely the extended period of the clock seen following exposure to MLN4924 is via Wnt inhibition but that rather it is due to the elevated levels of NICD brought about via inhibition of NICD degradation.

Reviewer 2:

*1) The experimental work nicely shows a correlation between NICD levels and stability and period length. It is however not clear if there is a causal relationship between elevated NICD half-life and extension of the oscillation period. The last set of experiments presented (subsection “Inhibition of SCF (SKP1- CUL1-F-box protein) E3 ubiquitin ligase complexes results in delayed clock oscillations and higher levels of NICD”), aiming at inhibiting the SCF E3 ubiquitin ligase complex is extremely important to address this issue. However, it should be extended to show that the inhibitor does not work through inhibition of the Wnt pathway, as is the case of the previous inhibitors. Otherwise the order of events remains unclear. Related to this, the last panel of*
Figure 6*, which is not described in the Results section (and should be), suggests that levels of phosphorylated β catenin are elevated following MLN4924 treatment. This should be explained*.

See response above to reviewer 1, point 5.

*2) In many ways,*
Figure 1
*is essential to this manuscript. We feel it could be enhanced to facilitate the reading of the manuscript and understanding of the model and predictions.*
Figure 1
*are not visually explicit enough to understand: a) the relationship between the clock protein, NICD, the feedback loops and the output on the oscillation period; b) how this model differs from the previous ones (Figure A could be replaced by the previous models); c) what the assumptions underlying this new model are. In addition, and maybe in the Experimental Procedures section, a statement explaining how the model can fit both the chicken and mouse somitogenesis, which oscillations for instance have a different period, would be useful*.

Figure 1 have been improved in order to provide a better graphical representation of the molecular oscillator. The diagrams have been annotated to provide insight into how changing model parameters affects model outputs. We welcome the suggestion of replacing Figure A but we wanted to illustrate to the reader that the release of NICD is a multicellular phenomenon that we are simplifying in this model. We have added an extended discussion of how the model is related to other models in the Discussion section.

We have simplified the model structure so that we can focus specifically on half-lives of activators/inhibitors of transcription. The model assumptions are explained in greater detail when the model is introduced.

The questions of how chicken-clock period is different to that of mouse is a very interesting one. Moreover, it is not properly understood how even in a single PSM the period of individual oscillators varies as a function of axial position!

However, this work does show that modifying the half-life of positive activators of clock genes can produce a shift in period that is of similar magnitude to the chick-mouse difference. Of course, this could be an important mechanism in setting species-specific period and controlling the oscillation rate along the axis but we do not currently have the data to address this question so have not commented upon it.

*3)*
Figure 1
*should be explained in a bit more detail, since this is the basis for understanding the rationale behind the experiments. The period on the top of the plots C-E correspond to what? In D and E how much are the half-lives of NICD and Hes7 increased? Maybe it should be stated clearly what the used values are and that the loss of oscillations in*
Figure 1
*corresponds to red region in*
Figure 1
*near the solid line that is the limit of oscillations*.

We thank the referee for pointing out the lack of detail. The period is the period of the oscillations for a given parameter set. Parameter values are now presented in the legend. The oscillating/no oscillating regions in parameter space are now better annotated.

*4) The competition relationship between NICD and the inhibitory clock protein to activate and repress transcription of a clock gene respectively (what parameter/properties considered?) impinges on the predictions of the model and interpretation of*
Figure 1*, which postulates that elevated NICD half-life results in more clock protein, and should be better spelled out*.

Upon introducing the model we state clearly that the competition between inhibitor/activator is a key model feature. The production function f is now introduced up front and we discuss that the assumption is that the balance of activator and inhibitor set the production rate.

*In addition, depending on the ability of the clock protein to be a potent transcription inhibitor, having NICD longer around but not much more NICD protein may not have much of an impact. It would be useful to explicit which parameter(s) in the mathematical model reflect this aspect*.

We have conducted a range of numerical experiments to classify model behaviour. What we suggest in this study is that the balance between activator/inhibitor is crucial. So making the half-life of inhibitor too large will result in loss of oscillations. This is now made clear for example in Figure 1 (increasing inhibitor half-life results in damped oscillations).

*5) The description of the approach to measure the half-life is a bit confusing and hard to follow with regard to the equations.* A_i_(t) *is the amount of NICD in both pools A and*
***B****. “t” is a free parameter. We suggest for the second equation in the subsection “Measuring the half-life of endogeneous NICD”: B_bar(τ)=A_bar(t+τ)=sum… and F(τ)=A_bar(t+τ)/A_bar(τ). This is more coherent with the integral form of F(τ) in the fourth equation*.

*Maybe it would be useful to have a profile sketch for A as a function of t. If we understand correctly, as long as τ<*t_s_*, A is equal to a constant value A*_*0*_*, then as we reach to* t_s_*, A(t) decays exponentially. Using this sketch, it will be easier to understand that for τ<=*t_s_*, A(t)=A(t+τ), else A(t+τ)=A(t)e*^*–*^*β*^*τ*^*). We also suggest that the first sentence of the second paragraph changes to “Letting* A_i_(t) *be the amount of NICD in the* i^th^
*tissue sample at time (*t*). We define the experimentally measurable sum in pool A as…”*

We thank the referee for these queries. We apologise as there was typo in the first equation and variables were not defined clearly enough. These issues have now been addressed.

There are nine different PSMs. Each are split in two with a half from each going into Pool A and Pool B. We define *A*_*i*_*(t)* to be the amount of NICD in the *i*^*th*^ sample in Pool A and *B*_*i*_*(t)* to be the amount of NICD in the *i*^*th*^ sample in Pool B. As pool B is sampled for *τ*  extra minutes we can write *B*_*i*_*(t)* = *A*_*i*_*(t + τ*). Summing over all the samples we get to the experimentally measurable quantity and from that we can infer the half-life.

6) What is T in the integral form of F(τ) in the subsection “Measuring the half-life of endogeneous NICD”, the clock period?

T is the period of the clock. This has now been amended. Apologies for any confusion caused by this omission.

*7)*
Figure 6*: It should be mentioned that the vertical axis is log F(τ), the units of the horizontal axis is minutes. The label seems to be “τ” and not “t”. It would be useful to add at the end of the legend (A) that the inverse of the slope gives the half-life (data presented in*
Figure 6*). I don't understand where does the 0.5C*_*0*_*=C*_*0*_*e-*^*slope t*^
*relationship come from. In the main text, Log F(τ)=–β(Τ–*t_s_*). From the slopes of*
Figure 6
*we get the values in*
Figure 6
*and from the intercept we should be able to get* t_s_
*(which seems to be of the order of 10 minutes). Can the authors comment on this (and why this has not been used to calculate* t_s_*?)*.

We thank the reviewer for pointing out these errors. The labels should be τ. This has been changed. The *0.5C*_*0*_*=C*_*0*_*e-*^*slope t*^ has been removed and we now refer to the main text for the correct formula.

The referee is correct that *t*_*s*_ takes a value of the order of tens of minutes. As stated in the definition of F(\τ), the measure ought to be necessarily noisy in the first few minutes when NICD is still potentially being produced. Owing to experimental restrictions NICD levels were samples at later times and it is from these data that we infer the half-life. Hence we are in the regime where t>*t*_*s*_.

*8)*
Figure 4
*and Figure 6 seem to point to different values for NICD half-life. The 2nd lane in*
Figure 4
*indicates that all NICD proteins are cleared in 3 hours; the 4th lane indicates that significant residual NICD remains 1h after its production has been blocked. Could the authors explicit how these experimental results fit with the calculated half-life indicated in the subsection “Exposure to XAV939, and the CDK inhibitors increases the half-life of cNICD in the PSM”?*

Figure 4 shows different exposures of NICD for the WB panels, chosen to best highlight the difference between NICD levels in ‘+’ and ‘–’ corresponding pools. For calculating the half-life of NICD following treatments the WB images of all the time points were taken at the same exposure for calculation.

*In addition, it would be very informative to visualize (in embryos) if the NICD half-life is homogenously increased in the somites and presomitic mesoderm, as its impact on the oscillation period is expected in a very defined spatial window*.

Despite our very best efforts to secure this data it has eluded us and we have found the antibody particularly tricky to work with by immunocytochemistry. We have put a huge amount of effort, time and reagents into optimising the protocol by either paraffin section, cryostat section and whole mount immunocytochemistry on explants but to no avail. A clear reliable signal is attained only very rarely. For this particular experiment the difficulty is compounded by the fact that we need to obtain a clear strong signal in pairs of explants to glean any information (two half-embryos from the same embryos where one is treated “+” and one treated “–” the inhibitor).

*9) Impact of SHH inhibition on the oscillation. It is a bit unsettling to only find in the Discussion a reference to the 2010 PNAS paper claiming the opposite results, and not upfront in the Results section. In addition, one would have liked to read possible reasons for the discrepancy*.

We mention the Resende paper in subsections “Wnt inhibition delays the pace of *cLfng* oscillations in the chick and mouse PSM” and “Shh inhibition does not delay the pace of *cLfng* oscillations in the chick PSM” and we have broadened our Discussion.